# Rational programming of history-dependent logic in cellular populations

Ana Zúñiga[1,6], Sarah Guiziou[1,5,6], Pauline Mayonove[1], Zachary Ben Meriem[2], Miguel Camacho [1], Violaine Moreau[1], Luca Ciandrini [1,3], Pascal Hersen[2,4] & Jerome Bonnet [1✉]

Genetic programs operating in a history-dependent fashion are ubiquitous in nature and govern sophisticated processes such as development and differentiation. The ability to systematically and predictably encode such programs would advance the engineering of synthetic organisms and ecosystems with rich signal processing abilities. Here we implement robust, scalable history-dependent programs by distributing the computational labor across a cellular population. Our design is based on standardized recombinase-driven DNA scaffolds expressing different genes according to the order of occurrence of inputs. These multicellular computing systems are highly modular, do not require cell-cell communication channels, and any program can be built by differential composition of strains containing well-characterized logic scaffolds. We developed automated workflows that researchers can use to streamline program design and optimization. We anticipate that the history-dependent programs presented here will support many applications using cellular populations for material engineering, biomanufacturing and healthcare.

[1] Centre de Biochimie Structurale (CBS), INSERM U154, CNRS UMR5048, University of Montpellier, Montpellier, France. [2] Laboratoire Matière et Systèmes Complexes, UMR 7057 CNRS & Université Paris Diderot, 10 rue Alice Domon et Léonie Duquet, 75013 Paris, France. [3] Laboratoire Charles Coulomb (L2C), University of Montpellier & CNRS, Montpellier, France. [4] Laboratoire Physico Chimie Curie, UMR168, Institut Curie, Paris, France. [5] Present address: Department of Biology, University of Washington, Seattle, WA 98195, USA. [6] These authors contributed equally: Ana Zúñiga, Sarah Guiziou. ✉email: jerome.bonnet@inserm.fr

Living organisms execute sophisticated tasks and generate highly complex structures at all scales[1–3]. A key feature enabling these rich behaviors is the ability of biological systems to modulate their response depending on the order in which signals are received. Such history-dependent responses are ubiquitous in biology from animal behavior down to the heart of fundamental processes like cell division (checkpoints), differentiation (cell-fate commitment), and development[4–6]. History-dependent behavior might also be important for microbial survival strategies by providing a fitness advantage in the evolutionary competition[7,8].

From a research and engineering perspective, the ability to reliably program living cells to produce history-dependent responses has appealing practical implications. These programs could be used as temporal and spatial trackers for decoding biological processes such as development. Furthermore, living organisms could be engineered to exhibit sophisticated behaviors not found in nature. History-dependent programs could be applied in biomanufacturing to mediate sequential, substrate triggered activation of components of a metabolic pathway. Bacterial therapeutics could be activated only after having encountered certain body locations in a particular order and reached a high density inside their target tissue, possibly in combination with external therapeutic triggers[9–12]. History-dependent morphogenetic programs are also pivotal to engineer synthetic tissues[13] and living functional materials that require order-specific, sequential assembly[14–17].

Scientists have started to program history-dependent gene expression in living cells, mostly using feedback mechanisms and site-specific recombination to record and store transient events, and transmit this information across cellular generations[18–25]. Compared to feedback-based systems, recombinase memory does not require constant protein production to hold state, reducing metabolic burden and increasing evolutionary stability of the engineered systems[26,27].

Site-specific recombinases permanently invert or excise DNA sequences flanked by a pair of target sites[19,21]. The state of the system can be encoded both by gene expression and within the architecture of the target DNA, which is modified via recombination. Recombinase switches are analog-to-digital converters and have been engineered to encode complex Boolean logic using reduced, single-layer architectures[28,29]. Stochastic, mutually exclusive recombination reactions have been used to randomly generate mosaics of reporter gene expression and enable cell lineage tracking in various organisms[30,31]. Researchers also built recombinase devices to control gene expression according to the number of inputs received by the cells and the order in which they appear.

Friedland et al. designed a recombinase cascade consisting of a daisy-chain of DNA memory modules, each composed of a recombinase gene flanked by its corresponding target sites. In response to the input, the recombinase inverts and turns OFF its own gene, while priming the following module for the next input by correctly reorienting an inducible promoter. Using this device, the authors programmed cells to express a reporter after a unique sequence of three inputs. However, this architecture lacks flexibility, and can only implement a fraction of all possible geneexpression programs.

Ham et al. proposed a different strategy in which recombinase target sites are interleaved, creating dependencies between recombination reactions. In this scheme, a recombinase can invert or excise another enzyme's target site. Thus, the target DNA transitions through different states according to the order in which recombination reactions occurred[27,30]. By inserting genes and regulatory elements at strategic positions of the DNA scaffold, gene expression is triggered in specific states. Roquet et al. refined this concept and built a DNA register capable of recording the occurrence and the order of up to three inputs. To do so, they used multiple pairs of orthogonal target sites for each enzyme, so that each state has a different DNA architecture and is distinguishable by sequencing. Roquet et al. then generated a database of registers containing different combinations and permutations of gene-expression elements, and simulated their behavior. Then, a few registers implementing specific two and three-input history-dependent programs were built and validated experimentally.

However, compared to other genetic design workflow[32,33], this method remains a trial and error process, and the ultimate functionality of the circuit is hardly predictable. Each program is executed using a different architecture, and context effects arising from unexpected parts interactions can alter recombination efficiency and gene expression. We previously reported such context effects for recombinase devices, as some attB and attP sites have cryptic promoter or terminator activities[28,33,34]. In addition, concatenating highly repetitive orthogonal sites[35] may impact the scalability of this strategy due to nonspecific recombination[31,36] and genetic instability[37].

Here, we present an alternative strategy for recombinaseoperated history-dependent programs. We aimed to deliver a systematic framework supporting the implementation of all possible programs, and satisfying the following design specifications. First, for a given number of inputs, a standard DNA scaffold should allow gene expression in any desired recombination states corresponding to the presence of inputs in a specific order. DNA scaffold standardization supports in-depth optimization and removal of context effects, providing recombinase devices with reliable behavior[29,33]. Second, orthogonal recombination sites should be avoided. Third, researchers should have access to readily usable software tools for automated design and optimization of history-dependent programs.

In order to meet these requirements, we turned to multicellular computation, in which the computational labor is distributed between different strains of a multicellular system. While each single-cell program is implemented using a different architecture[32,35], distributed programs are executed by combining different strains executing simple functions. The compartmentalization of circuit elements allows for thorough optimization and reuse of biological parts. Using this scheme, a small library of standard strains supports the implementation of a large amount of programs. Smaller circuit size in each strain also reduces metabolic burden. Finally, because of its modularity, multicellular computing is a powerful approach to systematize circuit design, and obtain a predictable behavior from engineered biological systems[38]. We and others have used multicellular computation to reliably implement a high number of Boolean logic functions using a small collection of wellcharacterized genetic devices[33,38,39].

Here, we engineered multicellular systems capable of executing history-dependent programs. We designed modular, standard DNA scaffolds in which genes can be inserted at specific locations that become transcriptionally active in different recombination states. Scaffolds allow the implementation in one strain of all possible gene-expression programs for a specific sequence of inputs. Programs requiring responses for different sequences of inputs are implemented by composing multiple strains. Each strain independently executes a portion of the whole program, without requiring cell–cell communication channels, thereby reducing the need for optimizing communication channels between cells. We automated circuit design and provide the community an easy to use, web-based design tool that generates DNA architectures and sequences for any history-dependent program.

In addition, to streamline scaffold optimization, we created a method termed Optimization via Synthesis of Intermediate Recombination States (OSIRiS) in which the different intermediate recombination states of the scaffold are generated in silico, synthesized, and tested. Using OSIRiS, we rapidly assessed scaffold function in all states and removed context effects altering gene expression.

Using this integrated workflow, we demonstrate the reliable execution of two- and three-input history-dependent programs by engineered multicellular systems. Strains can be composed in a combinatorial manner according to the desired program. Because of its high modularity, reliability, and automated design workflow, our approach offers an attractive alternative to systematically implement history-dependent cellular programs in living systems. We anticipate that the framework presented here will support many applications with intricate signal processing requirements in the fields of material engineering, biomanufacturing, and healthcare.

## Results

**A modular scaffold design to implement history-dependent programs.** History-dependent gene-expression programs can be represented as a lineage tree[40] in which each branch, or lineage, corresponds to a specific order of occurrence of the inputs. The number of lineages is equal to $N!$, where $N$ is the number of inputs (Fig. 1a). For instance, for 2-input programs, two lineages exist, and for 3-input programs, six lineages exist. We implemented history-dependent programs by using site-specific recombinases that perform DNA inversion and excision events (Fig. 1b); we focused on serine integrases that operate in several species and for which many orthogonal enzymes have been characterized and used to build genetic circuits[28,29,33,35,41].

We designed a modular scaffold capable of executing all possible 2-input history-dependent gene-expression programs that can occur within a single lineage (Fig. 1c). The modular scaffold contains insertion sites or "slots" in which genes of interest (GOI) can be inserted and expressed in each state of the lineage tree (Fig. 1c). Each input is assigned to an integrase that controls recombination of specific portions of the scaffold. Using this scheme, any possible combination of gene-expression states within a particular lineage can be achieved by simply inserting the desired gene at the corresponding positions, and switching the identity of recombination sites according to the desired lineage (Fig. 1d).

Depending on the identity of the different GOIs, the scaffold can express one or multiple genes across the different output states (Fig. 1e). Programs requiring gene expression in different lineages are decomposed into subprograms that are executed by different strains, each strain corresponding to one lineage. The full program is executed by a multicellular system obtained by mixing strains in equal proportions (Fig. 1e). Importantly, as our system does not use cell–cell communication, if one of the subprograms is ON, the global output of the system is considered to be ON. For a given number of inputs, the maximum number of strains needed is equal to the number of lineages ($N!$ for $N$ inputs) (Fig. S1a). However, most functions are implementable with fewer than the maximum number of strains, as the number of strains depends on the number of lineages in which gene expression is required (Fig. S1b).

Based on the same principle, we designed 3-, 4-, and 5-input scaffolds (Fig. S2). Each scaffold is derived from the previous one; new target sites and a gene slot are added to enable the detection of an additional input and the expression of an additional gene. These scaffolds support the implementation of all history-dependent programs up to 5 inputs.

**Automated design of history-dependent programs.** Automated design frameworks[29,32,42] that lower the entry barrier into novel technologies have proven to be critical in empowering a larger community with technological advances, providing unexpected innovations that could not have been envisioned by the original inventors. In this context, the automation of genetic circuit design is an important step toward the deployment of cellular computing systems into myriad research or engineering applications.

To automate the design of history-dependent programs, we encoded an algorithm taking a lineage tree as input (equivalent to a sequential truth table) and providing the biological implementation as output (Fig. S3), see "Methods" (https://github.com/synthetic-biology-group-cbs-montpellier/calin). The biological implementation consists of a graphical representation of the genetic circuit and the device DNA sequence of each strain (Fig. 1f). To enable broad access to our design framework, we provide a website called composable asynchronous logic using integrase networks (CALIN) (http://synbio.cbs.cnrs.fr/calin/sequential_input.php) that supports automated design of history-dependent programs. In the CALIN web interface, the user fills in the number of inputs to process and the desired lineage tree. The interface provides as an output the DNA architectures of the computational devices, the connection map between inputs and integrases along with the corresponding DNA sequences optimized for *E. coli*, which can be directly synthesized to obtain a functional system. The design of all history-dependent programs to up to five inputs and the maximum number of outputs corresponding to the number of states are accessible through this interface. The CALIN interface allows the scientific and engineering community to build upon our work to address many different problems.

**Implementing single-lineage programs.** We then aimed to experimentally implement our designs, starting with single-lineage, 2-input programs (Fig. 1c, d). Such programs could be used, for instance, to track if a cell has entered a specific differentiation pathway. In order to streamline the engineering process, we took advantage of the similarity of recombinase systems with state machines and designed an optimization workflow called OSIRiS (Fig. 2a). Because each state corresponds to a physically different DNA molecule in which sequence can be fully predicted, OSIRiS enables the different recombination intermediate sequences to be generated in silico, synthesized, and characterized (Fig. 2a). OSIRiS therefore decouples scaffold optimization from integrase-mediated switching, simplifies and shortens the characterization process, and allows for rapid iteration cycles. The OSIRiS script, written in python, automatically generates the intermediate sequences of any integrase-based scaffold, and is available on GitHub (https://github.com/synthetic-biology-group-cbs-montpellier/OSIRiS) and Codeocean[43].

We first used OSIRiS to generate all recombination intermediates for a 2-input, 3-output, single-lineage program (Fig. 2a). All recombination intermediates behaved as expected (Fig. 2a and Fig. S4). We then characterized the history-dependent response of nine single-lineage 2-input programs (Fig. 2c). We chose representative programs, based on their lineage, their number of ON states, and the number of different output genes. Single-lineage programs with single-output permits to identify particular states the cell entered, while those with multioutputs can enable precise, unambiguous discrimination between different states after a particular input order. We detected ON states using one or multiple fluorescent proteins. For one lineage (lineage 2: b then a), we tested all possible history-dependent programs producing a single type of output (green fluorescent protein (GFP)). For lineage 1 (a then b), we tested programs using a different fluorescent reporter for each state. We used a dual-controller

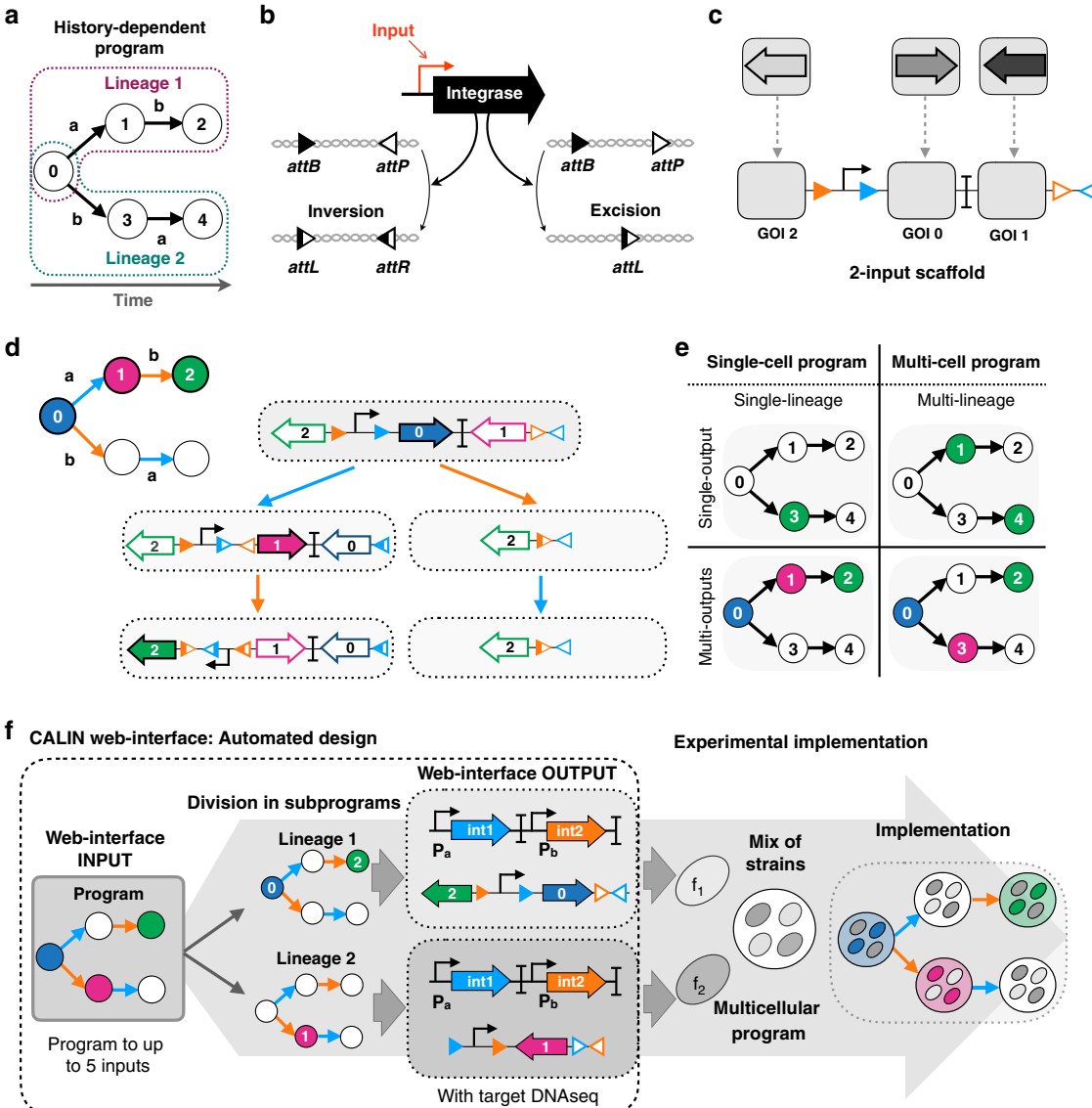

**Fig. 1 Design of a modular scaffold for 2-input history-dependent multicellular programs. a** Lineage tree representing a history-dependent program. Letters represent the presence of the two inputs (a and b) and numbers on nodes represent states of the system associated with the order of occurrence of the inputs. For two inputs programs, five states are possible. **b** Integrase-mediated excision or inversion. When integrase sites are in the opposite orientation (left panel), the DNA sequence flanked by the sites is inverted. If integrase sites are in the same orientation (right panel), the DNA sequence flanked by the sites is excised. **c** 2-input history-dependent scaffold. Integrase sites are positioned to trigger expression of an output gene (arrows) or not (empty gray squares) in the corresponding lineage. Programs are implemented by inserting genes corresponding to the ON states in adequate scaffold positions. **d** DNA transitions, recombination intermediates, and gene-expression states for the 2-input scaffold. The corresponding lineage tree is represented in the upper left. **e** Single-cell programs operate in a single lineage and can control expression of single or multiple outputs. Multicell programs operate in multiple lineages and can control the expression of a single or multiple outputs. **f** Automated design of history-dependent programs. The CALIN algorithm takes as input a history-dependent program written as a lineage tree. CALIN decomposes multilineage programs into subprograms, each corresponding to a different lineage (a then b; b then a). For each subprogram, the algorithm identifies the ON states and the order of inputs within the lineage. Based on this information, the biological design is computed, and the software provides input/integrases connections, the architecture of history-dependent scaffolds and their corresponding DNA sequences. Each subprogram is executed in a different strain as a DNA device ($f_1$, $f_2$). The full program is implemented by composing the different strains into multicellular systems.

plasmid[28] in which Bxb1 integrase is under the control of the pTet promoter responding to anhydrotetracycline (aTc, input a) and Tp901-1 integrase is under the control of the pBAD promoter responding to arabinose (Ara, input b) (Fig. 2b). For programs expressing GFP in multiple output states, we used different GFP variants: sfGFP, GFP226, and GFP221. Each of these variants has different fluorescence intensities and reduced sequence homology[44] to avoid genetic instabilities issues[32] (see Supplementary File). We operated the system in fundamental

mode, i.e., inputs cannot occur simultaneously, but only sequentially[45]. All nine single-lineage programs behaved as predicted (Fig. 2c). The scaffold was capable of driving expression of various fluorescent reporters in different DNA states and in both lineages. All devices had at least a tenfold change in fluorescence intensity between the OFF and ON states, with a maximum fold change of over 250 for sfGFP (Fig. S5, Supplementary Table 1). We observed, as expected, variations in fold changes depending on which GFP variant was used. We

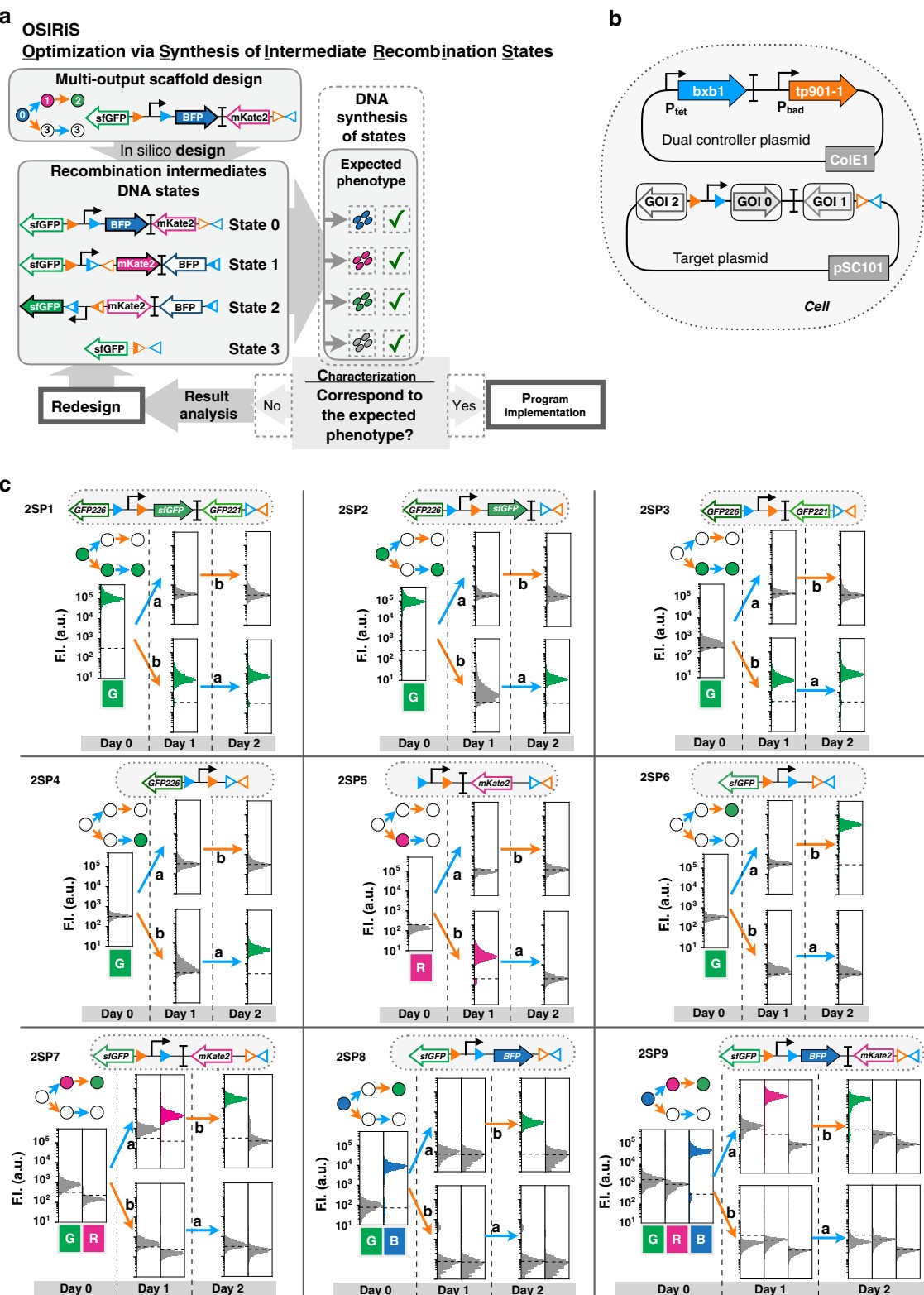

measured the percentage of switched cells, and found a minimum of 90% switching efficiency for states supposed to express a fluorescent protein. We also observed spontaneous switching in some states (mostly in S0) affecting a maximum of ~12% of the population (Fig. S5).

One particular program expresses a different fluorescent reporter gene in each input state of one lineage (Fig. 2c Fig. S4a, program 2SP9). We obtained the expected phenotype for each of

the four possible DNA states, BFP in state 0, RFP in state 1, GFP in state 2, and no expression in state 3 or 4 (Fig. 2c, Fig. S5). We analyzed the switching kinetics of the 2SP9 program by time-lapse microscopy, and observed a clear transition of states from blue to red to green fluorescence. We noted that 3 h were sufficient to observe 100% switching in the lineage states from S0 to S1 in strains growing continuously in a mother machine microfluidics device[46] (Fig. S6 and Supplementary Movie 1).

**Fig. 2 Characterization of 2-input single-lineage programs. a** OSIRiS workflow. A 2-input history-dependent scaffold with consecutive expressions of BFP, RFP, and GFP in one lineage is designed. The DNA sequences of the different input states corresponding to the intermediate recombination states are generated (DNA states). The DNA states sequences are synthesized and characterized and the phenotypes are compared to the expected ones. If they match, the implementation of the program is performed. Otherwise, the results are precisely analyzed to identify the origin of the failure, the multioutput scaffold is redesigned and a new OSIRiS cycle is performed. **b** genetic design of the two plasmid used to characterize each program. The dual-controller plasmid regulates the gene expression of the Bxb1 and Tp901-1 integrases. The target plasmid corresponds to the 2-input history-dependent scaffold. **c** Nine single-lineage history-dependent programs (2SP1–2SP9) exhibiting lineage specific, or state-specific gene expression with single or multiple genetic outputs were implemented and characterized. Bxb1 and Tp901-1 are induced by aTc (input a) and by arabinose (input b), respectively. The lineage tree for each program and its corresponding genetic DNA device are represented. Cells transformed with both plasmids were sequentially induced twice for 16 h, at which point fluorescence intensity was measured by flow cytometry. Each histogram shows the expression of fluorescent reporters expressed at different induction states. All experiments were performed in triplicate three times on three different days. A representative example from three biological replicates is depicted here. Fold change measurements can be found in Fig. S5.

**History-dependent logic in multicellular systems**. We then implemented 2-input history-dependent programs producing outputs in the two different lineages, therefore requiring the assembly of a multicellular system (Fig. 3a). As a demonstration, we built two multicellular programs, 2MP1 and 2MP2. To implement these programs we used four different strains, mixing two strains per program. We cocultivated the two strains for 3 days, performing overnight inductions with different sequences of inputs.

We measured the bulk fluorescence intensity of the cellular population in each state using flow cytometry and plate reader, and confirmed that programs behaved as predicted, with clearly measurable, successive expression of different genes depending on the order of inputs (Fig. 3b–e). Flow-cytometry and microscopy analysis showed, as expected, mixed gene-expression states (ON and OFF) between the two strains (Fig. S7). Flow-cytometry analysis of both multicellular systems showed an equivalent percentage (~50%) of cells in each subpopulation during 3 days of sequential induction, confirming the stability of the system and the lack of competition effects between the two strains (Fig. 3b, c, Fig. S7). Fluorescence intensity was still readable by plate reader, despite its decrease due to strain dilution within the multicellular system (Fig. 3d, e, Fig. S7).

We then sought to scale up the system, and designed a 3-input scaffold capable of expressing a different gene in every state of a 3-input lineage (Fig. 4a–c). We used OSIRiS workflow to optimize and characterize the 3-input scaffold and its five recombination intermediate states (Fig. 4d). We optimized the initial 3-input scaffold design to correct unexpected GFP fluorescence in states 0 and 4 (Fig. 4d). By removing two DNA spacer sequences (probably containing cryptic promoters activities) we obtained the expected behavior. The OSIRiS workflow proved extremely useful to accelerate the optimization process and obtain a functional version of the scaffold producing the expected output in all states (Fig. 5a, Fig. S8).

We then designed five single-lineage, 3-input programs and assessed their functionality in response to all possible combinations and sequences of inputs. We added a third integrase, Integrase 5 (Int5)[41] to the controller plasmid, under the control of the pBEN promoter responding to benzoate (input c) (Fig. 5b). We confirmed the functionality and quantified the recombination efficiency of each integrase in the triple controller plasmid by using OSIRiS intermediate DNA states as recombination targets (Fig. S9). Then, we cotransformed the triple controller with the corresponding target plasmid for each program, and performed sequential overnight inductions with aTc (Bxb1), arabinose (Tp901-1), and benzoate (Int5) for 4 days (see "Methods"). Measurements of fluorescence intensity by flow cytometry in each of the 16 input states were consistent with the expected program (Fig. 5c, Figs. S10 and S11). In addition, the fluorescence observed

for each input state were similar to those observed during OSIRiS characterization (Fig. 5a, c, Fig. S12).

We then composed various 3-input programs operating at the multicellular level. We assembled four programs (3MP1-4) with different fluorescent reporter genes expressed in different states in separate lineages (Fig. 6, Fig. S13). Bulk fluorescence intensities of all multicellular systems executing 3-input programs were measured by plate reader, and were in good agreement with the corresponding lineage trees (Fig. 6, Figs. S13 and S14). Flow-cytometry analysis was consistent with the expected multicellular behavior, with only one portion of the population expressing an output in a particular state (Fig. S15). Here, again, fluorescence intensity decreased proportionally with the dilution rate of each population in the mix, but was still measurable in bulk using a plate reader (Fig. 6, Figs. S13 and S14).

Some history-dependent programs also have a combinatorial logic component, and part of the program can be executed by a Boolean logic device (see Supplementary Text). We calculated that mixing Boolean and sequential logic devices can actually reduce the number of devices and the number of strains used (Fig. S16). We used previously designed Boolean integrase logic gates[33] to test this minimization strategy using a 3-input history-dependent program requiring four different strains in fully sequential mode (Fig. S17a). Combining Boolean and history-dependent devices, only three strains instead of four are needed to implement the same program (Fig. S17a). The measured fluorescence intensities in mixed populations were consistent with the expected lineage tree (Fig. S17b), confirming that Boolean-based minimization is indeed a viable approach to reduce multicellular system complexity. These results also highlight how the modularity and composability of distributed multicellular computation support the use of different families of logic devices within the same multicellular system.

We then scaled our scaffold designs and generated scaffolds for 4- and 5-input history-dependent gene-expression programs (Fig. S18). The 4-input scaffold allows for expression of a different GOI in each state of a given lineage, while the 5-input scaffold allows expression of a different GOI in each state except in the state 0 (with no input) (Fig. S18b). An additional strain is needed if gene expression is required in this state.

Taken together, these data demonstrate that multi-input/multioutput history-dependent programs can be reliably implemented in a distributed fashion across a cellular population, without the use of cell–cell communication.

**Robustness of history-dependent programs**. We then evaluated the fidelity of all implemented history-dependent programs using a vector proximity framework[29]. We calculated the similarity between the biological data for history-dependent programs and their ideal implementation (see "Methods") (Fig. 7). In this representation, the lower the angle deviation is from 0°, the closer

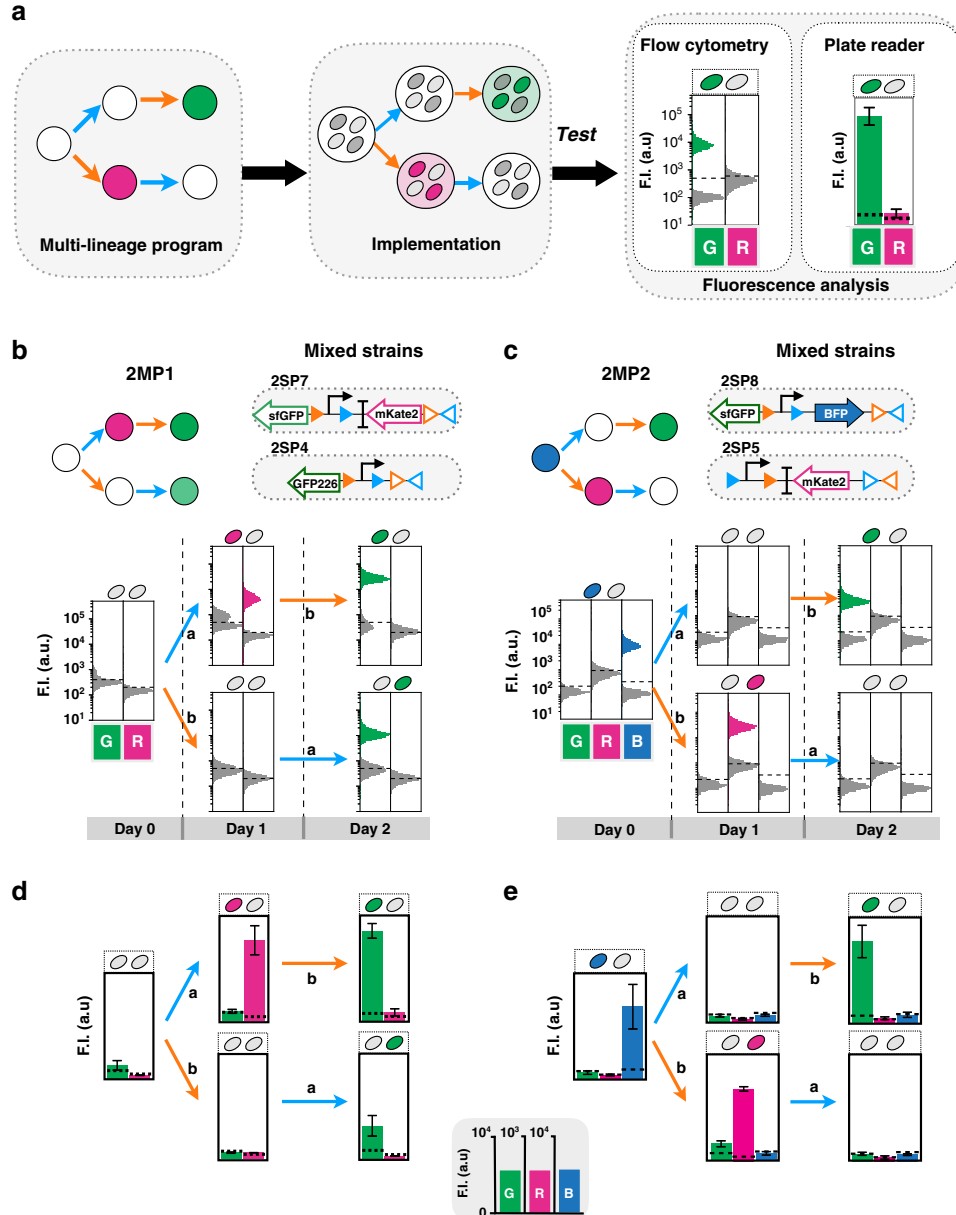

**Fig. 3 Characterization of 2-input multicellular programs.** Workflow characterization of 2-input multicellular programs (**a**). After the input program design, single-lineage programs are mixed and the multicellular program is implemented. The program characterization is done by fluorescence measurements by flow cytometry and plate reader. The flow-cytometry analysis allows us to observe the percentage of population ON and OFF for one state. Design and characterization of 2-input multicellular program 2MP1 and 2MP2, by flow cytometry (**b**, **c**) and plate reader (**d**, **e**), respectively. Both multicellular programs were implemented using two different single-lineage strains. The lineage trees for each program and its corresponding genetic DNA device are represented. The inputs are represented by letters, a for aTc inducing Bxb1 Integrase and b for arabinose inducing Tp901-1 integrase. To implement each program, the strains were mixed in similar proportions, grown for 16 h and sequentially induced with each molecule. Each histogram shows the expression of fluorescent reporters at different induction states. A representative example is depicted here. The bar graph corresponds to the mean value of the fluorescence intensity (F.I.) in arbitrary units (a.u) for each fluorescent channel (G (GFP), R (RFP), and B (BFP)) with linear different scales. All experiments were performed in triplicate three times on 3 different days (data distribution in dot plots in Fig. S7c–f). The error bars correspond to the ±standard deviation of the mean of the three different experiments. The dotted line indicates the negative autofluorescence from control strain. Note that GFP226 has a lower fluorescence intensity than sfGFP, as expected.

the program behavior is to its expected one. For instance, a program exhibiting a behavior opposite to the expected one would have an angle of 90°. Because a high recombination efficiency is essential to obtain a good program implementation, we evaluated the global recombination efficiency across the whole program using the percentage of the cells switched in each state of the program. We call this value "switching rate" and we considered programs with similarity angles smaller than 5° in

excellent agreement with the expected outcome; programs with angles between 5 and 20° were qualified as reliable, while the ones with angles higher than 20° were not recommendable to use.

We found that single-lineage devices had a very robust switching behavior for both 2- and 3-input programs (Fig. 7a), mostly with angles lower or equal to 5° (11/14) and none resulted in an angle of more than 15°. Because all programs had a high switching rate, we extended our analysis to program output

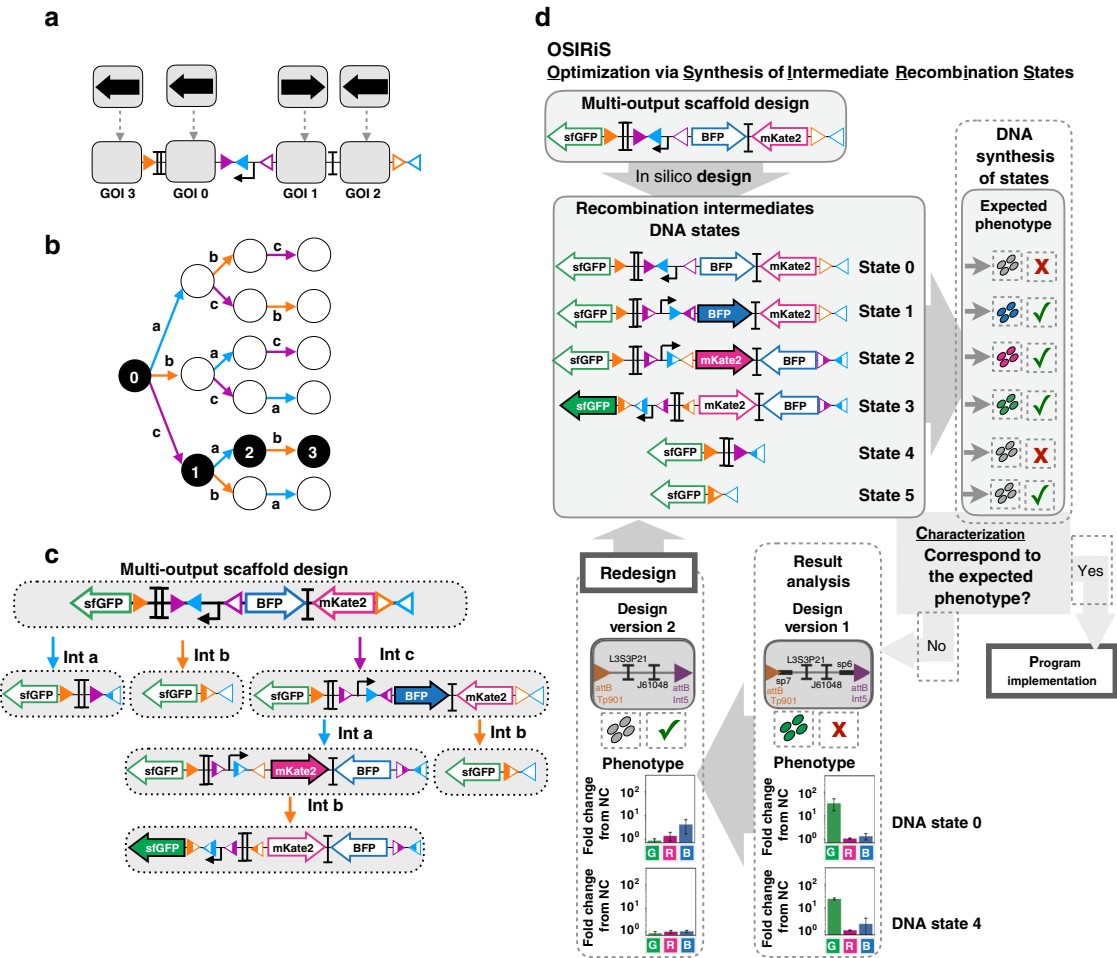

**Fig. 4 Design of a modular scaffold for 3-input history-dependent programs.** 3-input history-dependent scaffold (**a**) and its lineage tree (**b**). Integrase sites are positioned to permit expression of an output gene in various states of the lineage tree. For each state of the desired lineage, a different gene is expressed, and a gene is also expressed when no input is present. The four columns of the lineage tree correspond to different numbers of inputs that have occurred sequentially (from 0 to 3 inputs) and the six lineages correspond to different order of occurrences of inputs (example: a–b–c for lineage 1 and b–a–c for lineage 3). **c** DNA and gene-expression states of the scaffold. The gene at the GOI position 0 is expressed only when no input is present. The scaffold has six different DNA states. **d** Optimization of the 3-input scaffold using OSIRiS. For a given 3-input program, a scaffold with consecutive expressions of BFP, RFP, and GFP in one lineage is designed. From this design, six intermediate DNA states are generated and the expected phenotype for each DNA state of the tree is predicted. Two versions of this 3-input scaffold were analyzed. Version 1 was producing unexpected GFP fluorescence in state 0 and state 4. Version 2 is an optimized design from version 1, in which two DNA sequences corresponding to spacers sp7 and sp6, flanking L3S2P21 and J61048 terminators, were removed. The fluorescence intensities in different channels for two versions in DNA states 0 and 4 is shown. The bar graph corresponds to the fold change over the negative control (strain without fluorescent protein) for each channel (GFP, RFP, and BFP) from three experiments with three replicates per experiment. The error bars correspond to the standard deviation between the fold changes obtained in three separate experiments.

intensities, and monitored the similarity between the experimentally measured fluorescence intensities and the theoretical ones. The robustness of fluorescence was also close to the expected, with angles lower than 17° (Fig. S19). Only the program 3SP3 presented a higher angle (30°) and lower minimal fold change of BFP fluorescence (Fig. S19, Table S1), presumably because of a lower BFP expression, and not because of a lack of robustness of the switch (Fig. 7a). In addition, we evaluated the robustness of the implemented multicellular programs by measuring the similarity between expected and experimental fluorescence intensity of the system. We plotted these angles versus the average fold change in fluorescence intensity for each program. We found that the six multicellular programs operated reliably, with angles between 8° to 12° (Fig. 7b). As expected, multicellular programs had fluorescence fold change lower in average than other programs, due to dilution effect. In general, programs

behaviors are highly similar to expected ones, demonstrating the predictability of the modular scaffold operation.

## Discussion

Implementing rich behavior in living organisms requires the development of robust frameworks for history-dependent logic[47,48]. In this work, we demonstrate that complex history-dependent programs can be executed in a simple manner by distributing the computational labor between different members of a multicellular system. Multicellular computation provides highly modular and composable systems, i.e., many logic functions can be implemented in a straightforward manner from a reduced set of strains executing basic functions. Here, we designed, characterized, and optimized 2- and 3-input scaffolds. We built nine 2-input and five 3-input single-lineage logic

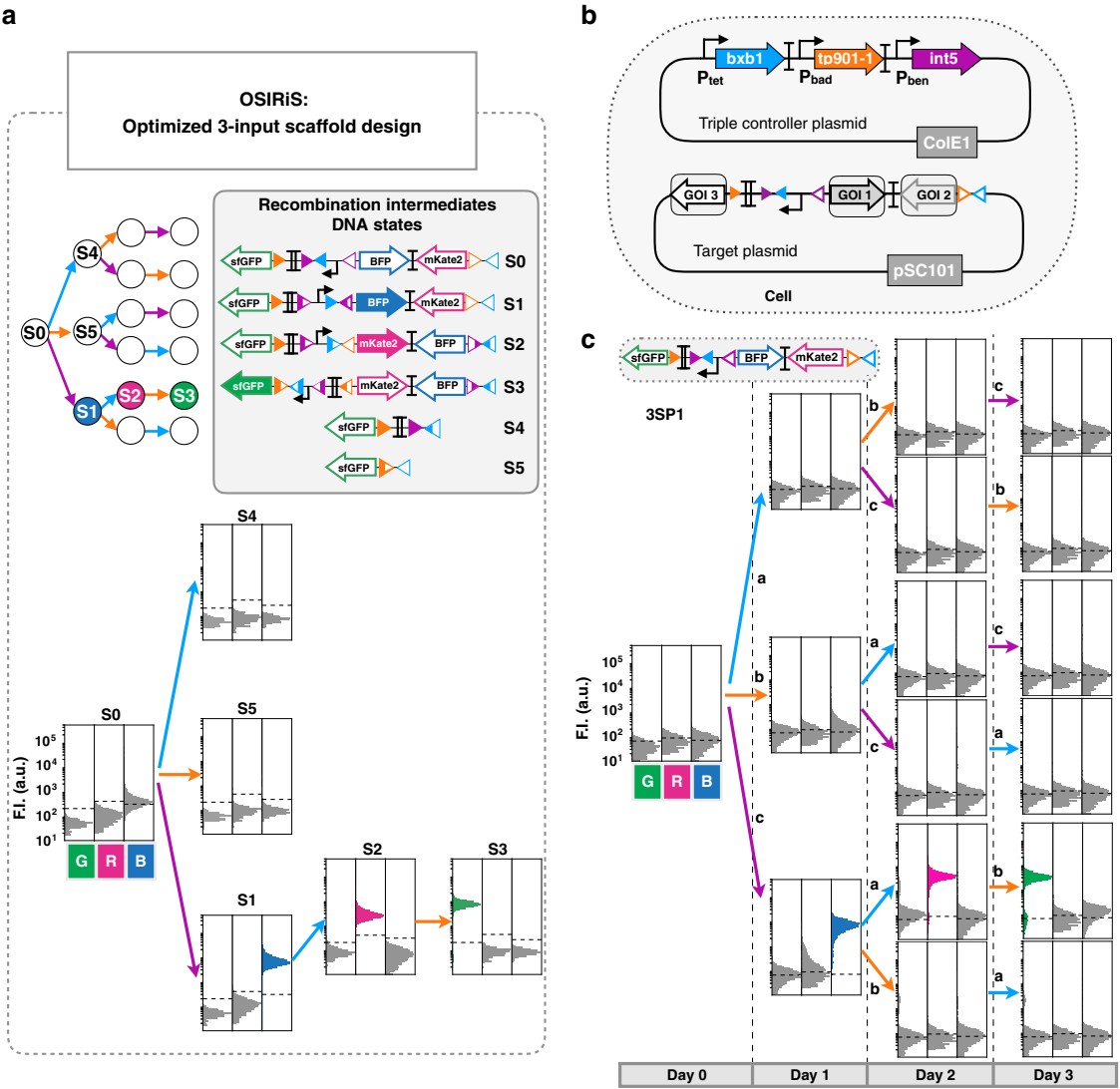

**Fig. 5 Characterization of scaffold for 3-input history-dependent programs. a** Characterization of the final 3-input scaffold and its recombination intermediates DNA states by flow cytometry. We characterized each DNA state by measurement of GFP, RFP, and BFP fluorescence intensities. Each histogram shows fluorescent reporters expressed in the different DNA states, from three experiments with three replicates per experiment. A representative example is depicted here. A detailed design for the final 3-input scaffold and fold change measurements can be found in Fig. S8. **b** Genetic design of the two plasmids used to implement 3-input programs. The triple controller plasmid regulates the expression of Bxb1, TP901-1, and Int5 integrases. The target plasmid corresponds to the 3-input history-dependent scaffold. **c** Characterization of a 3-input history-dependent scaffold. We cotransformed the 3-input program with the triple controller plasmid. Bxb1 expression is induced by aTc (input a), Tp901 by arabinose (input b), and Int5 by benzoate (input c). The lineage tree for the program and its corresponding genetic DNA device is represented. For characterizing the system, cells were sequentially induced three times for 16 h each, with different order of occurrences of inputs. Each histogram shows fluorescent reporters expressed in different states. All experiments were performed in triplicate three times on three different days. A representative example is depicted here. Fold change measurements can be found in Fig. S12.

devices. All devices had predictable behavior close to the ideal program implementation. Without further optimization, we composed various strains containing single-lineage devices to implement complex history-dependent programs executed at the multicellular level, without the need of communication channels. The fact that our system is built by composing well-characterized, standardized genetic elements allowed us to automate program design in a straightforward manner. We provide an easy-to-use web interface that gives, from a desired program, the corresponding circuit designs and DNA sequences.

Here, we successfully implemented 21 history-dependent programs, and found that all programs showed a high robustness of execution, even after three successive inputs. A key parameter influencing the robustness of history-dependent programs is the

recombination efficiency of the integrases used. If recombination efficiency is too low, incomplete switching leads to error propagation across a sequence of inputs. In our case, we found that the robustness of the system was mostly affected by TP901-1 enzyme which has a 92% recombination efficiency, while Bxb1 and Int5 had efficiencies close to 97%. Obtaining high recombination efficiencies by optimizing the integrase generator or the target devices will thus be critical to avoid the compounding effect of incomplete recombination when implementing 4- and 5-input programs.

Using the single-lineage programs from this study, 36 different programs can be executed. Here, we demonstrated that in one program, three different states can be unambiguously distinguished using different fluorescent proteins (Figs. 2c, 3, and

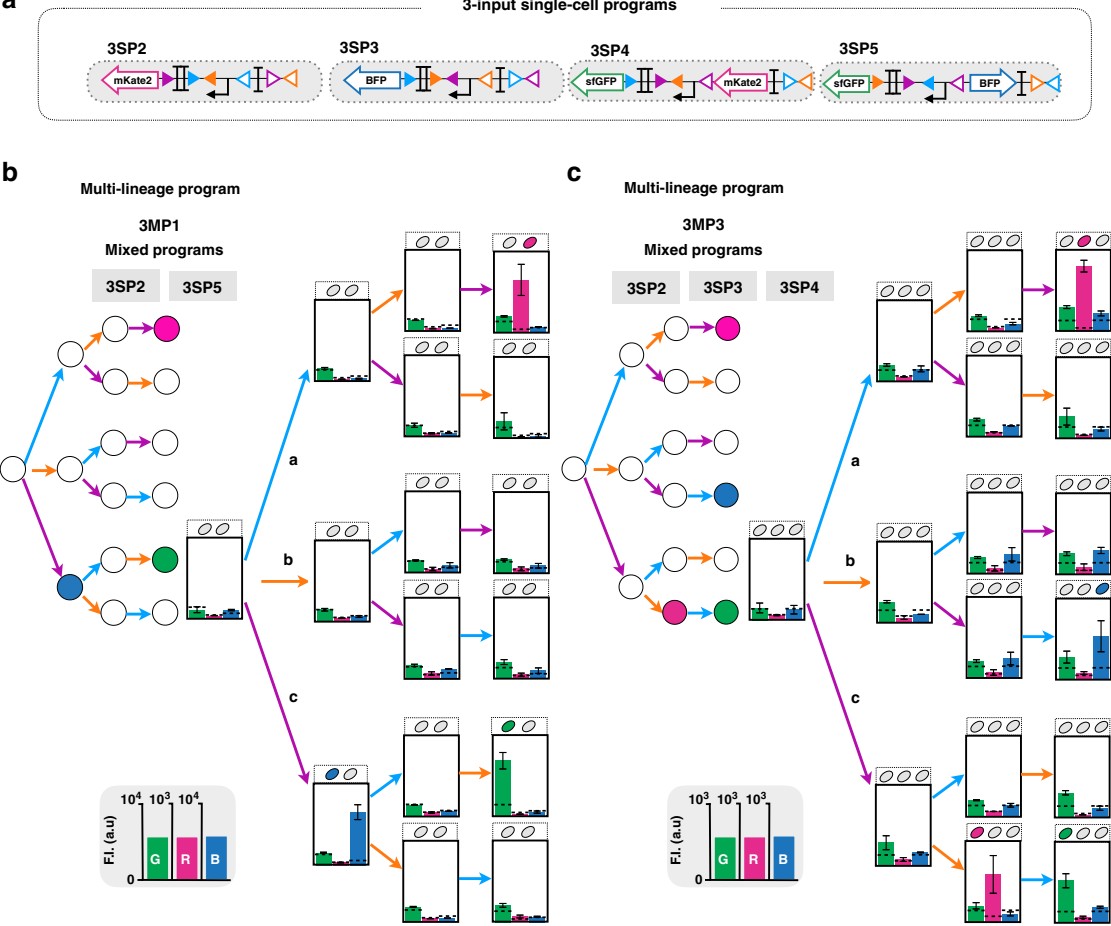

**Fig. 6 Characterization of 3-input multicellular programs.** Design and characterization of 3-input single-cell (single-lineage) programs operating in various lineages (**a**). Multicellular programs were composed by mixing two (**b**) or three (**c**) single-lineage programs. The inputs are represented by letters, a (blue) for aTc (expression of Bxb1 Integrase), b (orange) for arabinose (expression of integrase TP901-1), and c (purple) for benzoate (expression of Int5). Strains were mixed in equal proportions and sequentially induced three times for 16 h each, with different order of occurrences of inputs. The bar graph corresponds to the mean value of fluorescence intensity (F.I.) in arbitrary units (a.u) for each fluorescent channel (GFP, RFP, and BFP), with different and linear scales each. All experiments were performed in triplicate three times on three different days (data distribution in dot plots in Fig. S14). The error bars correspond to the ± standard deviation of the mean of the three different experiments performed in triplicate on three different days, and measured using a plate reader. The dotted line indicates the autofluorescence of negative control strain.

5c). More states could be distinguished by fluorescence microscopy by fusing fluorescent proteins to subcellular localization tags[49], expressing various combinations of fluorescent proteins and performing spectral deconvolution[50], or expressing barcoded mRNA and detect them through smFISH.[31]

Because intermediate recombination states are encoded within DNA, we developed a rapid prototyping workflow (OSIRiS) based on total synthesis and characterization of all recombination intermediates. We found that the results from scaffold characterization using OSIRiS closely matched device behavior when operating in real conditions, i.e., responding to recombinase mediated inversions or excisions. The recombination intermediate DNA states provided by OSIRiS can be used to optimize the scaffold and characterize the recombinase efficiencies of the integrases in the genetic context and experimental condition in which the program will be used (Fig. S9).

While we do operate our system in fundamental mode, some inputs could arrive in a mixed manner. The time resolution of the system (i.e., the minimum time between two inputs so that different outcomes can be observed) and the proportion between the resulting populations having entered different lineages would depend on the dynamics of the recombination reactions, which is

governed by integrase expression, stability, and catalytic rate, among others. In this context, our system could be used to reverse engineer the timing between inputs, as already shown before[51].

One potential challenge for application requiring a detectable readout like biosensing is that the system output is produced by a single strain at a time. However, our systems with up to three strains still have a well-detectable output, similarly to previously engineered Boolean recombinase logic systems operating at the multicellular level[33]. If needed, cell–cell communication could be used to propagate the output state to all the strains of the population[42]. DNA-sequencing methods could also be used to address the state of the system[27,35,52,53]. Finally, for many applications like morphogenetic engineering, strain specific phenotypic output would be desirable. By simply changing the output gene of interest, various phenotypes like secretion, adhesion, or motility could be implemented in an history-dependent fashion.

Another challenge is the number of strains required to implement programs requiring gene expression in multiple lineages. For example, a maximum of six strains is required to execute all 3-input programs. Importantly, we showed here and in previous work (Fig. S7, Figs. S13 and S14 and ref. [33]) that multicellular systems operating for several days exhibited reasonable

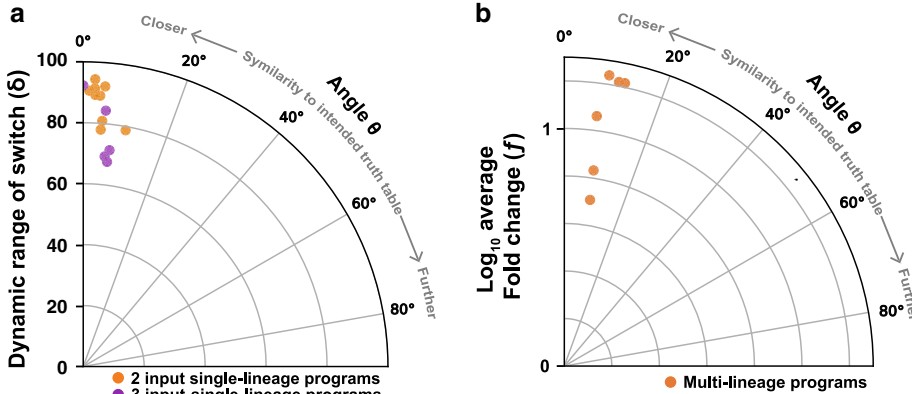

**Fig. 7 Robustness of history-dependent programs.** Angles representing the similarity between biological and the ideal implementation of history-dependent programs were plotted. The lower the angle deviation is from 0°, the closer the program behavior is to its expected one. **a** Switching robustness, with angles computed from percentage of cell versus the worst-case switching rate of the switch ($\delta$), were plotted. **b** Implemented multicellular programs robustness with angles evaluated between biological and expected fluorescence from plate reader measurement versus the logarithm of average fluorescence fold change ($f$) were plotted.

stability. If needed, synthetic cooperative behavior could be used to maintain all strains in the population[54,55]. Strains could also be cultivated in separate chambers within a microfluidic device for instance, avoiding competition problems[56]. Alternatively, the number of strains required to implement complex logic circuits can be reduced by combining Boolean[33] and history-dependent devices. Future research exploring minimization strategies and alternative scaffold designs could help reduce even more the number of required strains.

The functionality of our system could be expanded in several manners. For instance, integrases mediate irreversible switching, and similarly to other recombinase devices, our system can respond only once to a given input. The addition of recombination directionality factor[26] would result in history-dependent programs with reversible state transitions and increased complexity. Cell–cell communication could also provide another layer of complexity to coordinate synthetic collective behavior[57].

Because of the high modularity of integrase logic, the system presented here could be quickly repurposed by swapping the signals controlling integrase expression to detect chemical (e.g., metabolites, biomarkers of disease, pollutants) or physical signals (e.g., temperature, light). Here, we show using the benzoic acid sensor BenR that a viable approach to engineer new switches is to start by optimizing the properties of the transcriptional controller and then screen for different integrase variants with different translational efficiency or stability (Fig. S20). As serine integrases work in many species, including mammals and plants, the history-dependent programs presented here could be used in myriad research and engineering applications. Programs responding to environmental signals could enable therapeutic cells to act in a highly precise spatiotemporal fashion, increasing specificity and therapeutic efficacy while reducing side effects. Cellular sensors could perform temporal logic analysis for diagnostics, environmental, or manufacturing applications. Finally, sequential programs would support iterative, ordered assembly of synthetic biological architectures for tissue and biomaterial engineering[58,59].

## Methods

**Strains, media, and inducers**. *E. coli* strain DH5alphaZ1 (laci$^q$, PN25-tetR, Sp$^R$, deoR, supE44, Delta(lacZYA-argFV169), Phi80 lacZDeltaM15, hsdR17(rK−, mK+), recA1, endA1, gyrA96, thi-1, relA1) was used in experimental measurements of history-dependent programs. DH5alphaZ1 was grown on LB media with antibiotics corresponding to the transformed plasmid(s) to do the cloning. For experimental measurements the cells were grown in Azure Hi-Def medium (Teknova, 3H5000)

supplemented with 0.4% of glycerol. Antibiotics used were kanamycin 25 μg/mL and chloramphenicol 25 μg/mL. The inducers were: L-arabinose (Sigma-Aldrich, A3256) used at a final concentration of 0.7% wt/vol; anhydrotetracycline (Sigma-Aldrich, 37919) used at a final concentration of 20 ng/mL and benzoic acid (Sigma-Aldrich, 242381) used at a final concentration of 100 μM, for 3-input programs.

**Integrase controller and target plasmids construction**. One-step isothermal Gibson assembly was used[60] to build all plasmids described. Vectors pSB4K5 and J64100 (from parts.igem.org) were used to construct all genetic circuits. The pSB4K5 vector containing kanR cassette and pSC101 origin of replication was used to clone different input programs, based on DNA scaffold design, including BP and LR targets, GOI, and other DNA parts. The J64100 plasmid containing camRY cassette and ColE1 origin of replication was used to clone the integrase controller cassette. Enzymes for the one-step isothermal assembly were purchased from New England Biolabs (NEB, Ipswich, MA, USA). PCR were performed using Q5 PCR master mix and One-Taq quick load master mix for colony PCR (NEB), primers were purchased from IDT (Louvain, Belgium), and DNA fragments from Twist Bioscience. Plasmid extraction and DNA purification were performed using kits from Biosentec (Toulouse, France). Sequencing was realized by GATC Biotech (Cologne, Germany).

To build the triple controller plasmid, we constructed an inducible cassette containing the *benR* gene regulator, constitutively expressed by promoter J23106, the P$_{ben}$ promoter (both *benR* and P$_{ben}$ from metabolic and sensing modules, respectively)[61], and Int5[41] controlled by P$_{ben}$ promoter (see vector map). The cassette was inserted upstream of dual controller integrase plasmid[26]. After assembly the triple controller vector was transformed and cloned in *E. coli* strain DH5alphaZ1.

The design of DNA scaffold for target plasmid was done using a python script to automatically generate a library of DNA sequences (https://github.com/synthetic-biology-group-cbs-montpellier/Generate_DNAseq), which minimizes the number of errors. Moreover, because the final sequences result from permutations of a reduced set of parts, Python is particularly well suited for the task. All sequences were designed to support cloning by Gibson assembly at an identical location in pSB4K5 template vector. Consequently, all sequences were composed of the 40 bp spacer 0 at 5′ end, and 40 bp spacer N at 3′ end. The DNA sequences for every designed program were synthesized, as linear fragments, by Twist Bioscience. Each DNA fragment was PCR amplified and assembled between spacer 0 and N in pSB4K5 template vector. All DNA sequences of history-dependent programs are listed in Table for Supplementary Information. Target plasmids were transformed and cloned in *E. coli* strain DH5alphaZ1. All plasmids were purified using QIAprep spin Miniprep kit (Qiagen) and sequence verified by Sanger sequencing in Eurofins Genomics, EU.

**Experimental conditions and sequential induction assays**. Integrase controller and target plasmids were cotransformed in *E. coli* strain DH5alphaZ1 and plated on LB agar medium containing kanamycin and chloramphenicol antibiotics. Three different colonies for each program to test were picked and inoculated, separately, into 500 μL of Azure Hi-Def medium (Teknova, 3H5000) supplemented with 0.4% of glycerol, kanamycin, and chloramphenicol in 96 DeepWell polystyrene plates (Thermo Fisher Scientific, 278606) sealed with AeraSeal film (Sigma-Aldrich, A9224-50EA) and incubated at 30 °C for 16 h with shaking (300 rpm) and 80% of humidity in a Kuhner LT-X (Lab-Therm) incubator shaker. All experiments were performed in the same condition of growth. After overnight growth the cells were

diluted 1000 times into fresh medium with antibiotics and let them grow at 37 °C for 16 h (day 0). For multicellular programs, cell strains harboring different programs were equally mixed, before diluting (1000-fold total dilution) into fresh medium with antibiotics. The mixed populations were grown at 37 °C for 16 h (day 0). After 16 h of incubation, the cells were serial diluted (1000-fold total dilution), first 10 µL culture into 190 µL of fresh medium in presence of antibiotics and a second dilution (from first dilution) 10 µL into fresh media with antibiotics and inducers (aTc, arabinose or benzoate) or not. The cells were grown at 37 °C for 16 h (day 1). After overnight incubation, the cells were serial diluted (1000-fold total dilution), first 10 µL culture into 190 µL of fresh medium in presence of antibiotics and a second dilution (from first dilution) 10 µL into fresh media with antibiotics and inducers (aTc, arabinose or benzoate) or not. (day 2, aTc → Ara; Ara → aTc, for 2-input programs and aTc → Ara; aTc → benzoate; Ara → aTc; Ara → benzoate; benzoate → aTc; benzoate → Ara, for 3-input programs). For 3-input programs, after overnight incubation the cells were serial diluted (1000-fold total dilution), first 10 µL culture into 190 µL of fresh medium in presence of antibiotics and a second dilution (from first dilution) 10 µL into fresh media with antibiotics and inducers (aTc, arabinose or benzoate) or not. (Day 3, aTc → Ara → benzoate; aTc → benzoate → Ara; Ara → aTc → benzoate; Ara → benzoate → aTc; benzoate → aTc → Ara; benzoate → Ara → aTc.) To measure cell fluorescence, aliquot of cells from each day were diluted 200 times into Attune Focusing Fluid (Thermo Fisher Scientific, A-24904) and incubated for 1 h at room temperature before flow-cytometry analysis. For plate reader measurement an aliquot of cells from each day were diluted four times in phosphate buffered saline PBS before read. For microscopy analysis, aliquots of cells from each day were mixed with glycerol at final concentration 15% v/v and kept at 80 °C until its analysis. Every experiment was repeated two or three times.

**OSIRiS design and characterization**. To automate the design of intermediate recombination state, we developed a python script giving from any integrase-based scaffold and a list of integrase of interest the recombination intermediate DNA sequences. OSIRiS code is widely available on GitHub (https://github.com/synthetic-biology-group-cbs-montpellier/OSIRiS), requires Python 2.7 and Biopython installation.

To use OSIRiS script, a csv file containing integrase sites of interest is required, we provided a csv file containing widely used integrase sites but any specific sequences can be added. Of note, this script is only considering irreversible DNA excision and inversion. As output, GenBank file with integrase site annotations is obtained for all intermediate recombination states.

In our scaffold design, we selected the fluorescent proteins sfGFP, mKate2, and BFP, as their excitation and emission spectrums do not overlap. We used P6 as the promoter and B0034 as the ribosome binding site[62]. To insulate the translation from the genetic context, we placed a ribozyme in the 5′ end of each output gene, catalyzing the cleavage of the mRNA at this position[63]. We used different ribozymes for each output gene (RiboJ, BydvJ, and AraJ) to avoid multiple repetitions of sequences in the construct. Based on this design, we generated intermediate recombination input states using our OSIRiS python script. We then synthesized and constructed these sequences. We characterized all the constructs by flow cytometry. The fold change over the negative control was determined from mean value over that of the negative control. The mean fold change was represented in the figure corresponding to the mean of the fold change of the three different experiments.

**Flow-cytometry and plate reader analysis**. Flow cytometry was performed on Attune NxT flow cytometer (Thermo Fisher) equipped with an autosampler and Attune NxT™ Version 2.7 Software and BD LSR Fortessa (Becton Dickinson), with FACSDiVa software. Experiment on Attune NxT were performed in 96-well plates with setting; FSC: 200 V, SSC: 380 V, green intensity BL1: 460 V (488 nm laser and a 510/10 nm filter), and red intensity YL2: 460 V (561 nm laser and a 615/25 nm filter). Setting for experiments on Fortessa were FSC: 400 V, SSC: 300 V, green intensity GFP: 580 V (488 nm laser and a 530/30 nm filter), red intensity mCherry: 565 V (600 nm laser and a 610/20 nm filter), and blue intensity V1: 460 V (405 nm laser and a 450/50 nm filter). All events were collected with a cutoff of 20,000 events. Every experiment included a positive control expressing GFP, RFP, or BFP and a negative control harboring the plasmid but without reporter gene, to generate the gates. The cells were gated based on forward and side scatter graphs and events on single-cell gates were selected and analyzed, to remove debris from the analysis (Fig. S21), by Flow-Jo (Treestar, Inc) software. Original FCS files for each program are available on Flow repository at the links provided in the "Data availability" section.

Plate reader measurements were done on Cytation 3 microplate reader (Biotek Instruments, Inc) and data were collected using Biotek Gen 5 software. After induction time, cultures were diluted four times in PBS and measured with the following parameters: GFP: excitation 485 nm, emission 528 nm, gain 80, BFP: excitation 402 nm, emission 457 nm, gain 70, RFP: excitation 555 nm, emission 584 nm, gain 100, absorbance: 600 nm. For each sample, GFP, BFP, and RFP fluorescence intensities were normalized to absorbance at 600 nm. The arbitrary units of fluorescence were used to graph bars in each figure. Plots for all data analysis were obtained using python plotting codes available on GitHub[64]. Plate reader data are available as a Supplementary Excel file.

**Microscopy and microfluidic analysis**. Cell samples from sequential induction experiments were analyzed by confocal microscopy. Images were acquired using a Zeiss Axioimager Z2 apotome, Andor's Zyla 4.2 sCMOS camera (MRI platform, Montpellier). Two microliters of cells were spotted on Azure/glycerol 2% agarose pads. Images were taken from phase contrast, GFP, RFP, and BFP fluorescence images at ×100 magnification. Images were analyzed using OMERO software. Source images are provided as a Source Data file.

For microfluidic analysis, experiments were performed using a mother machine microfluidic device consisting in arrays of parallel chambers (1 µm × 1 µm × 25 µm) connected to a large channel. Chambers were fabricated using electron-beam lithography on SU-8 photoresist (MicroChem), while the channel was fabricated using soft-lithography. From the subsequent master wafer, microfluidic chips were molded in polydimethylsiloxane (PDMS) and bonded to a glass slide using plasma activation. Cells, grown overnight in LB supplemented with Cam and Kan, were then loaded into the chambers by centrifugation on a spin coater using a dedicated 3D printed device. LB media flown in the mother machine are supplemented with Cam and Kan, but also with 5 g l$^{-1}$ F-127 pluronic to passivate the PDMS surfaces and prevent cell adhesion. The medium diffuses to the chambers, providing nutrients and chemicals of interest to cells. Chemical inducers (aTc at 200 ng/mL and arabinose at 1%) were added to the media as required using solenoid valves (The Lee Company). A peristaltic pump was used to flow the various mediums through the device at a flow rate of 90 µL/min. Both the microfluidic device and the medium were constantly held at 37 °C. Images were obtained using an inverted Olympus IX83 microscope with a ×60 objective. Fluorescence levels were measured within a small rectangular region of interest located at the top of each chamber where a single cell is trapped.

**Automated generation of genetic designs**. Equations for the determination of number of subprograms to implement in different strains for history-dependent logic

History-dependent programs are represented as a lineage tree. Each node of the tree corresponds to a specific state of the system in response to a different scenario: when no input occurred, when one input occurred, and when multiple inputs occurred in a particular sequence. For an $N$-input program, the number $\sigma$ of states is equal to

$$\sigma = \sum_{k=0}^{N} \frac{N!}{k!}. \tag{1}$$

Then, for $N$-input/1-output history-dependent logic programs, the number $P_1$ of possible programs is equal to

$$P_1 = 2^\sigma = 2^{\sum_{k=0}^{N} \frac{N!}{k!}}, \tag{2}$$

as all states can have either an ON or OFF output. Similarly, for $N$-input/$M$-output history-dependent logic programs

$$P_M = (M+1)^\sigma = (M+1)^{\sum_{k=0}^{N} \frac{M!}{k!}} \tag{3}$$

programs exist. The maximum number of outputs (Fig. S22) is equal to the number of states as theoretically a different gene can be expressed in each state, then the maximum number of N-input history-dependent logic programs is equal to

$$P_{Max} = (\sigma+1)^\sigma = (\sum_{k=0}^{N} \frac{N!}{k!} + 1)^{\sum_{k=0}^{N} \frac{N!}{k!}}. \tag{4}$$

The maximum number of strains needed to implement an $N$-input/$M$-output history-dependent gene-expression program is equal to $N!$, which corresponds to the number of possible lineages in an $N$-input lineage tree.

**Automated generation of genetic designs to execute multicellular history-dependent gene-expression programs**. We encoded an algorithm capable of creating up to 5-input history-dependent program design using Python (Fig. S3) (https://github.com/synthetic-biology-group-cbs-montpellier/calin; http://synbio.cbs.cnrs.fr/calin/sequential_input.php). The algorithm takes as input a lineage tree (equivalent to a sequential truth table). The output corresponds to the biological implementation, such as a graphical representation of the genetic circuit and its associated DNA sequences for each strain. The lineage tree is decomposed into subtrees consisting of a single-lineage containing one or multiple ON states. This decomposition is done by iteratively subtracting the lineages containing ON states. To obtain the lowest number of subprograms, we prioritize among the lineages with ON states the ones for which the highest number of inputs occurred (from the right to the left of the lineage tree). After decomposition, for each selected lineage, two pieces of information are extracted. First, based on which states are ON, we directly design the corresponding scaffold by specifically inserting genes at the adequate GOI positions. Second, the order of occurrence of inputs corresponding to the lineage is used to identify which sensor modules are needed among the different connection possibilities between control signals and integrases. Then, by combining the design of the different lineages, we obtain the global design for biological implementation of the desired history-dependent gene-expression

program. To simplify the construction process of logic circuits, DNA sequence of computation devices is generated by our Python code (available also at Code Ocean platform). In CALIN, sequences are adapted for *E.coli*. But sequence generation can be adapted to other organisms (databases are available for *Bacillus subtilis* and *Saccharomyces cerevisiae*) or customly designed using the source Python code available on GitHub (https://github.com/synthetic-biology-group-cbs-montpellier/calin).

**Robustness analysis**. We adapted and generalized the vector proximity method described in Weinberg et al.[29] in order to compare the outputs of the programs to the desired theoretical behavior.

Each implemented program has a different number of inputs (up to three) and outputs; the matrix $T$ corresponding to the expected outcomes presents a number of rows equal to the number of states $\sigma$, Eq. (1), and three columns maximum (the RGB output channels). This matrix can be mapped to a $3\sigma$-dimensional vector $t$ by stacking all rows of $T$ one on top of the others. Similarly, we constructed a vector $s$ from the table $S$ of the experimental outcome (percentage of switched cells or fluorescence values for each channel and state). From the definitions of the scalar product

$$s \cdot t = \sum_{i=1}^{3\sigma} s_i t_i = |s||t|\cos\theta, \qquad (5)$$

we can compute the angle $\theta$ between $s$ and $t$ as

$$\theta = \cos^{-1}\frac{\sum_{i=1}^{3\sigma} s_i t_i}{|s||t|}, \qquad (6)$$

where $s_i$ and $t_i$ are the $i$-th components of the vectors $s$ and $t$, and $|\cdot|$ represents the Euclidean norm. If $s$ and $t$ are proportional, i.e., if the experimental output is an ideal implementation of the program, the angle is 0°. The larger the angle, the worse the accuracy of the program with respect to the desired behavior.

As explained below, the table $T$ gathering the expected program's outcomes is computed differently when comparing the percentage of switch or fluorescence from a single strain, or when testing a program composed of many strains.

When analyzing the percentage of switch for each state from flow-cytometry data (Fig. 6a), the values in the signal table $S$ are bound by 1 (when all cells switch), and the table $T$ is the actual binary truth table. Instead, when comparing fluorescence values for single strains as in Fig. S18, the 1's of the table $T$ are replaced by the fluorescence of the positive control (measured by flow cytometry) of each corresponding reporter gene (referred to strains expressing constitutively the fluorescence reporter gene), and the 0's by the negative control (background values). This allowed us to consider fluorescence variation in programs with different GFP variants, and to avoid capping the signal values as done in Weinberg et al.[29].

In case of multicellular systems, the table $T$ is constructed by summing the fluorescence measured with the plate reader for each individual strain, then the resulting values are divided by the total number of strains used to implement the multicellular program (Fig. 7b). This procedure relies on the hypothesis that each strain equally contributes to the total observed fluorescence. The matrix $T$ gathering the expected outcomes can then be compared with the experimental output $S$ as previously explained.

We quantified the strength of the signal output by computing the dynamic range $\delta$ of switch, considering the absolute difference between the minimum percentage of cells in the ON input state and the maximum percentage of cells in the OFF input state (radial coordinate in Fig. 7a). In mathematical terms, we computed

$$\delta = \min_{i|t_i=1} s_i - \max_{i|t_i=0} s_i. \qquad (7)$$

When analyzing fluorescence values of individual strains (Fig. S18), we calculated the minimum fold change $f$ of each program considering the fold change between minimum fluorescence in the ON input state and maximum fluorescence in the OFF input state for each RGB channel

$$f = \left(\min_{\text{ON states}} s_i - \max_{\text{OFF states}} s_i\right)/F_{NC}, \qquad (8)$$

bearing in mind that the matrix $S$ in this case gathers fluorescence values and where $F_{NC}$ is the fluorescence of the negative control for the corresponding output channel. In the radial coordinate of Fig. 7b we plotted the $\text{Log}_{10} f$, meaning that a radial coordinate of 1 corresponds to a fold change $f = 10$.

Finally, we averaged the fold change of different channels when multiple strains are used in the implementation of multicellular programs and plot the logarithm of its value (Fig. 7b).

**Reporting summary**. Further information on research design is available in the Nature Research Reporting Summary linked to this article.

## Data availability

DNA sequences for all constructs are available as a Supplementary File. All raw data are available in Supplementary Files. Original FCS files are available on Flow repository[65];

2SP1; 2SP2; 2SP3; 2SP4; 2SP5; 2SP6; 2SP7; 2SP8; 2SP9; 3SP1; 3SP2; 3SP3; 3SP4; 3SP5; 2MP1; 2MP2; 3MP1; 3MP3. The webpage for CALIN is available at: http://synbio.cbs.cnrs.fr/calin/sequential_input.php. All other raw data are available from the corresponding author on reasonable request. Plasmids are available from Addgene: 2SP1 ID:126526; 2SP2 ID: 126527; 2SP3: 126528; 2SP4 ID 126529; 2SP5 ID: 126530; 2SP6 ID: 126531; 2SP7 ID: 126532; 2SP8 ID: 126533; 2SP9 ID: 126534; 3SP1 ID: 126535; 3SP2 ID: 126536; 3SP3 ID: 126537; 3SP4 ID: 126538; 3SP5 ID: 126539; and pITC (Integrase triple controller) ID: 126540. Source Data are provided with this paper.

## Code availability

All source codes are available on GitHub: https://github.com/synthetic-biology-groupcbs-montpellier/calin; https://github.com/synthetic-biology-group-cbs-montpellier/OSIRiS.

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

## Acknowledgements

We thank the synthetic-biology group and members of the CBS for fruitful discussions. We thank Montpellier Ressources Imagerie (MRI) for microscope and cytometry support, especially Marie-Pierre Blanchard and Myriam Boyer. Support was provided by an ERC Starting Grant "Compucell," the INSERM Atip-Avenir program and the Bettencourt-Schueller Foundation. S.G. was supported by a Ph.D. fellowship from the French Ministry of Research and the French Foundation for Medical Research (FRM) FDT20170437282. Z.B.M. and P.H. were supported by an ERC Consolidator grant "Smartcells." The CBS acknowledges support from the French Infrastructure for Integrated Structural Biology (FRISBI) ANR-10-INSB-05-01.

## Author contributions

S.G., A.Z., and J.B. designed the project. S.G., A.Z., Z.B.M., M.C., and P.M. performed the experiments. S.G., A.Z., Z.B.M., M.C., L.C., and P.H. analyzed data. S.G. wrote the python source code for CALIN and V.M. and S.G. implemented the web server for CALIN. S.G., A.Z., L.C., P.H., and J.B. wrote the manuscript.

## Competing interests

The authors declare no competing interests.
