## [Peer Review File · Nature Communications]

Reviewers' Comments:

Reviewer #1:

Remarks to the Author:

The manuscript by Zuniga et al describes a set of genetic constructs that can be modified by a number of different site-specific recombinases in an irreversible fashion to transition between a number of different sequence "states", with some of the states being associated with fluorescent expression and some not. The constructs are designed such that the sequence in which the recombinases are induced makes a difference for the expression of a fluorescent protein; for example if recombinase "a" is activated before recombinase "b", GFP is expressed; however if the order is reversed, GFP is not expressed.

Further, the authors make the system more complex by engineering plasmids that respond differently to the same sequence of recombinase activation; for example plasmid X will produce GFP when the order of activation is ab (but not ba); and plasmid Y will produce RFP when the order is ba (but not ab). By mixing the strain that harbor plasmids X and Y, and by observing the culture, the authors can figure out which sequence actually took place.

The paper is technically well done, in the sense that the plasmids and the constructs seem to work as expected from them, in the narrow technical sense. I do have some reservations however with respect to 1) the novelty of the work and 2) the applicability of the concepts to tasks in molecular logic. In the following I will expand on these concerns

1. Novelty.

The fact that a DNA backbone can be used as a state repository and manipulated by site-specific recombinases has been proposed more than 10 years ago and has since been confirmed experimentally in numerous works. The idea that a unique sequence of inputs can result in a specific output was already shown in 2009, Friedland et al, "Synthetic Gene Networks That Count". The concept was further explored in another publication by Roquet et al, "Synthetic recombinase-based state machines in living cells" in 2016. So while there are differences in the specifics of genetic design, the basic idea has been explored to saturation. The idea to split different plasmids between different strains has been proposed and implemented by the group of the lead author in two publications (refs. 37, 40). While an argument could be made that these references dealt with combinatorial logic in which the order of inputs was engineered not to matter, the technical difference in terms of construct design is minor and the main difference is in the "optics" and the interpretation of essentially the same technical approach.

2. Application to tasks in molecular logic

The authors choose a very special framework to interpret their transitions as "lineages". This implies that inputs are not applied more than once (and if they do, there is no effect because the irreversible switch acts the first time the recombinase is activated). Their lineages are really sequences of inputs of the following format: "first appearance of a happens before the first appearance of b, that happens before the first appearance of c...". In this sense, for N inputs there are N! sequences, as correctly stated by the authors; N! behaves exponentially for large Ns. With that, the number of outputs the authors can generate is not N! but rather on the order of N; moreover, for some sequences of inputs, the output is triggered in the middle of a sequence. So my question is, what exactly is the computational power of this approach? I agree that it can be engineering to generate a readout when a particular sequence of inputs is provided. This however was shown by Friedland et al. With strain mixing, an external observer can detect up to N different sequences out of N!, therefore, most sequences would result in no output. The extension to multiple strains does not strike me as a big conceptual leap; focusing on a single strain, the main result of the paper, if I rephrase it without undue embellishments, is as follows:

"For a pre-defined sequence of chemical inputs, one can engineer a DNA cassette that will generate GFP if and only if this particular sequence of inputs is provided to the strain harboring the cassette".

This would be a meaningful statement to make, however I could not find explicit reference to this result in the manuscript. Indeed, the examination of their three-input plasmids revealed that the same output could appear in different branches of the tree and at different time points. This is a serious problem, because it would mean that the same output represents different sequences of inputs, and thus the latter cannot be deduced from the output alone.

Computational design of 4- and 5- input sequences was described but the cassettes have not been tested.

To summarize, I think the paper could be of potential interest to Nature Communications but it has to be thoroughly rewritten, with the focus on the novelty and the computation power of the approach (what exactly does it compute? Does it always work within the boundaries? Are there "undetected" sequences?). The aspect of multiple strains working in parallel should be reduced to its correct proportions, as a trivial way to run a number of detectors, and not as some major conceptual breakthrough. The focus should be on a single unique sequence detector, on the generality of the design approach (eg does a unique solution exist for any arbitrary sequence of inputs, which will respond exclusively to this sequence), the scalability beyond 4 and 5 inputs. The already built in non-uniqueness, with the same output appearing in many different tree branches, casts serious doubts on the above raised points.

An experimental demonstration of 4- or 5-step sequence would certainly add to the paper.

The degree to which the authors would address these concerns will affect my eventual evaluation of the manuscript one way or the other. For now too many things are left open.

Reviewer #2:

Remarks to the Author:

In this paper Zúñiga et al. present a very interesting methodology for developing genetic circuits able to produce history-dependent responses. Despite this work can be of interest for the scientific community, there are several questions that must be addressed previous publication.

1. The method proposed is based in the use of recombinases to induce irreversible DNA modifications. Because experiments were done at population level, which is the efficiency of recombination? In other words, for a given population how many cells do not perform the proper DNA modification? This is a very relevant aspect because the sequential expression of different recombinases with different efficiencies can induce an accumulative effect. As a consequence, at the end of the experiment could be a distribution of populations expressing different genes. For simplicity, assuming that all recombinases have the same recombination efficiency p , after N inputs only p^N fraction of the final population will exhibit the correct behaviour. For instance, assuming that $p=0.85$, i.e. 85% of cells properly recombine, for systems responding to $N=4$ inputs only 52% of the cells would accumulate the right sequence of DNA modifications. Obviously, this effect will be more relevant for systems responding to higher number of inputs. Is this effected observed in the experiments? This point has to be addressed in the paper.

2. Related with the previous question, authors claim "Because all programs had a high switching rate, we extended...", However, "high switching rate" is not very precise, can these rate be quantified?

3. Authors claim "For a given number of inputs, the maximum number of strains needed is equal to the number of lineages ($N!$ for N inputs). However, most functions are implementable with fewer than the maximum number of strains, as the number of strains depends on the number of lineages in which gene expression is required." The expression "with fewer" is ambiguous. A chart representing the number of functions versus the number of strains for different number of inputs would be very useful.

4. Considering ON and OFF gene expression as different states, for N inputs should be $N! \cdot 2^N$ combinations, i.e. $N!$ branches with 2^N possible configurations of ON-OFF genes. Figure 2 shows nine different history-dependent responses to two inputs. Are these circuits arbitrary examples? Why these cases are selected?

5. Related with the previous question, creation of multicellular systems able to access to all $N! \cdot 2^N$ combinations could need more of N recombinases in each strain? Using 2^N recombinases all ON-OFF combinations are accessible, but probably this can be done with less recombinases. How the number of recombinases limits the subset of accessible functions?

6. Is there leakiness in the expression of the recombinases under inducible promoters? With RBS strength has been used to minimize leakiness? Could be useful to present genetic parts sequence as supplementary information.

7. How the uses of GFP variants instead using different reporters can provide confusing results? In particular, related with recombination efficiency, if populations subsets that have not properly recombined expresses a different GFP than cells that have recombined right, can this conceal this fact?

8. It is surprising the stability of mixed populations growth without observable competition effects. However, when the number of branches increase the number of different strains coexisting increases as well. Can this compromise the stability of circuits? For instance, for systems with 3 inputs could be necessary the coexistence of 6 different strains in the same culture, and scale up to 24 for 4 inputs. How this can compromise the applicability of this circuits for systems beyond 2 inputs?

9. According to the methodology, this system cannot be applied for temporal input patterns in which the same input appears more than one time, for instance in oscillatory patterns. How this aspect can limit the range of potential applications of the methodology presented? and can be possible to extend the approach to include patterns with repeated inputs? I sort discussion about this limitation can be useful

Response to reviewers: “Rational programming of history-dependent logic in cellular populations.”

Ana Zúñiga, Sarah Guiziou, Pauline Mayonove, Zachary Ben Meriem, Miguel Camacho, Violaine Moreau, Luca Ciandrini, Pascal Hersen, and Jerome Bonnet.

Montpellier, May 8^h 2020

We would like to thank the reviewers for their constructive comments which we believe have improved the quality of the paper. We have thoroughly rewritten the paper, in particular the introduction, to address comments, and make clear the novelty of our approach compared to previous work, and the advantages of distributed computation. Below is a full point-by-point response to the reviewer's comments. Our answers are in blue.

Reviewers' comments:

Reviewer #1 (Remarks to the Author):

The manuscript by Zuniga et al describes a set of genetic constructs that can be modified by a number of different site-specific recombinases in an irreversible fashion to transition between a number of different sequence "states", with some of the states being associated with fluorescent expression and some not. The constructs are designed such that the sequence in which the recombinases are induced makes a difference for the expression of a fluorescent protein; for example if recombinase "a" is activated before recombinase "b", GFP is expression; however if the order is reversed, GFP is not expressed.

Further, the authors make the system more complex by engineering plasmids that respond differently to the same sequence of recombinase activation; for example plasmid X will produce GFP when the order of activation is ab (but not ba); and plasmid Y will produce RFP when the order is ba (but not ab). By mixing the strain that harbors plasmids X and Y, and by observing the culture, the authors can figure out which sequence actually took place.

The paper is technically well done, in the sense that the plasmids and the constructs seem to work as expected from them, in the narrow technical sense. I do have some reservations however with respect to 1) the novelty of the work and 2) the applicability of the concepts to tasks in molecular logic. In the following I will expand on these concerns

1. Novelty.

The fact that a DNA backbone can be used as a state repository and manipulated by site-specific recombinases has been proposed more than 10 years ago and has since been confirmed experimentally in numerous works. The idea that a unique sequence of inputs can result in a specific output was already shown in 2009, Friedland et al, "Synthetic Gene Networks

That Count". The concept was further explored in another publication by Roquet et al, "Synthetic recombinase-based state machines in living cells" in 2016. So while there are differences in the specifics of genetic design, the basic idea has been explored to saturation.

Answer: We agree with the reviewer that we might not have sufficiently emphasized the original contributions of our work. In order to address this point, we have added some results in the main text and changed their presentation; we have also rewritten a great part of the introduction to better put our work in context.

Our changes cover two categories: 1) Highlighting the systematization and simplification of the engineering of history-dependent programs and our contribution to the community 2) Comparison with previous methods cited by the reviewers. Below we will discuss these points; we realize this is a long response, but we thought it was warranted to give the reviewer a satisfactory answer.

1) Systematization and simplification of the engineering of history-dependent programs, contribution to the community.

Besides the programs that are executed, a lot in this paper is about systematization in engineering, obtention of predictable program behavior, either related to system design (modular, standardized design and design automation) or debugging (quantification of robustness and rapid performance assay). We also strived to make our work accessible to the community, by providing the plasmids on addgene along with software tools that are easy to use even for non-experts. We are enabling scientists and engineers to build on our work for similar or other applications using recombinases. In that respect, our work goes well beyond what has been proposed before. More in details:

- A main, and still current challenge in the field of synthetic biology is the predictable assembly of systems that operate in a reliable fashion. Such goal implies quantifying the system's reliability, identifying failure modes, understanding the causes of failure, and finally correcting the problem. Here we designed an optimization method, OSIRiS, that allows us to rapidly measure the performance of the system in its different expected states, based on DNA synthesis. Using OSIRiS, we were able to identify states in which the system was malfunctioning and corrected the issue which was due to unexpected transcriptional activity of some of the biological parts we used (see new Fig. 4d). OSIRiS code is available on GitHub and CodeOcean.
- We quantified individual program robustness and compared them by adapting a method proposed by Weinberg et al (1)

- We precisely quantified systems behavior, either using a combination of flow cytometry, microscopy, or plate reader. All raw data are provided and are available on public repository.
- We designed a target DNA architecture that is highly modular, permits reuse of components, and contributes to the obtention of predictable behavior.
- Finally, we provide an open-source, easy to use software for automated design. We set-up a web-server, and provide all code on git-Hub and code ocean. In addition to providing the DNA architectures, our tool also provides DNA sequences for constructs. The GitHub version allows users to use parts for different organisms (*B. subtilis*, *S. cerevisiae*...) but also to use their own library of parts. This represents a leap forward for the design of recombinase devices.

We agree with the reviewer that these points were not necessarily put sufficiently forward in the previous version of the paper, and hence we decided to: (i) move the description of genetic design automation process earlier in the results section, and (ii) move the description of the OSIRIS method and its results in the main text (Fig. 2a, and Fig. 4d, and Fig. 5a). (iii) rewrite the intro to provide a better explanation of our goals, the challenges that we aimed to address in this work, and the design specifications we had. We believe that these changes will better highlight our original contribution to the field.

2) Comparison with previous methods for engineering history-dependent behavior in living cells cited by the reviewers.

Here also we agree that we could have done a better job in describing previous work, putting it in perspective, and highlighting the challenges we aimed to solve. We think we have done that in the new version of the paper.

Of course, several groups, including ours, have proposed and used recombinases to encode memory in living cells. Engineering is based on the reuse and improvement of previous concepts. Recombinase devices are no exceptions. Historically, the idea of using recombinases to build memory units can be traced back to 35 years ago (2). Ham and Arkin articulated in 2007 the idea of using recombinases to track orders of occurrence of inputs and control gene expression by having the DNA register to transition between different states (3). Yet the experimental implementation presented several issues (getting such devices to work reliably is not as trivial as one could expect).

Now, more specifically, the reviewer's cited two papers that built and experimentally tested recombinase-based devices that can produce an output only for a particular sequence and number of inputs (4, 5). In the following, we will discuss these works, their achievements, what we consider to be limitations. We will then highlight how our system differs, and how it provides

improvements for specific areas. Of course, the reviewers will understand that the goal here is not to discuss which system is the “best” (as this will depend on specific applications), but more to give a comparison, based on several criteria, and show that we provide advances that address some shortcomings of others implementations.

- a) **Friedland et al.** built one 3-input responsive device, in which a particular order of three inputs produces a final output (GFP). This is an original design, which can be compared to a DNA “abacus”. One device is demonstrated, and using this architecture, by swapping the different inducible promoters, it would be possible, for 3-inputs, to produce a final output for the $N!=6$ different possible sequences in a relatively straightforward manner. As it is, this architecture can implement 6 programs. However, for 3-inputs, there are actually 65536 possible history-dependent gene expression programs. In short, this system is not complete, and this design does not provide a general approach to implement all possible gene expression programs.

We could discuss how the highly intertwined SIMM architecture could be modified to make it complete. The coupling between recombinase expression and DNA inversion in the same genetic construct limits the modularity of the system. For example, it would be challenging to permanently express a reporter gene in a single state. For instance, for the sequence a, b, c; produce an output when a then b have occurred, but turn it off when c occurs. Of course, expression of the gene of interest could be coupled with the recombinase (in a polycistron for example), but in that case, its expression would be dependent on the presence of the input, which would somewhat diminish the utility of a memory device in the first place. In addition, the system is made to react to only one of the different possible input sequences.

In conclusion, while the Friedland et al. system remains an important milestone in the field, and can be perfectly suited for specific scenarios, this architecture is not modular and does not allow to control gene expression in various states. In other words, the scalability and systematization of this approach are limited.

- b) **Roquet et al.** first built a DNA register in which recombination sites are interleaved as in the work of Ham and Arkin, and Hsiao and al. (3, 6). They additionally use “mutant”, orthogonal pairs of recombination sites, i.e., sites recombined by the same enzyme but which cannot recombine between them.

The DNA scaffold changes its state at each input transition and each particular state can be identified by the DNA sequences which are different in each case. The authors then use their scaffold to construct gene expression programs in which expression of particular output genes is dependent on the order of inputs. The same scaffold is used as it is and the authors built a software to test different possible insertions of gene expression elements at different positions of the scaffold and their activation upon

different sequences of events. The authors experimentally implemented five 2-input programs, and two 3-input programs, of which one with a final output, and one with the same output constantly expressed in one of the six lineages is shown.

Again, this paper is a milestone in the field. Yet, there are several points that could be limiting in some cases.

- First, the use of mutant recombination sites, which are well known not to be completely orthogonal (7, 8). This could lead to scalability problems as the number of inputs increases.
- Second, because the DNA scaffold was not designed specifically for controlling gene-expression programs in various states, architectures are generated using a brute-force approach, then screened to identify specific programs. Consequently, for each program, a different DNA architecture is used. One can expect a high variability in program behavior as unpredictable context effects will certainly arise, and be different for each architecture. These facts could complicate program implementation, and serious optimization would be required to obtain a working system.

In all, it is not straightforward to know if all programs are actually implementable using this approach, and if one tries it would certainly require time-consuming optimization. In addition, because of the use of orthogonal sites, it is not clear if the system can be scaled.

Our work contrasts with these two papers on several points. We specifically designed a DNA scaffold to control gene expression in various states. Because we use a standard architecture, we provide highly reliable gene expression scaffolds allowing predictable implementation of all programs. We have thoroughly optimized scaffold operation. The distributed architecture facilitates program composition. Our system is complete, and is designed to implement all possible programs. We consider that for 3-input, the several programs demonstrated with different lineages, states, and reporters give us enough confidence that all programs can be realized. One challenge of our approach can be that for some programs we require the use of a high-number of strains, but many programs can operate within the number of strains we tested, and other strategies could be used (these are described in the discussion of the paper, and include synthetic cooperation to maintain ecosystem diversity (9, 10) or physical separation (11).

We have modified the introduction and discussion to put our work in a better perspective, especially compared to the two articles cited by the reviewer.

We also provide a comparison table between different history-dependent systems, as material for reviewers (see annex of this document). We're happy to add it to the supplementary material to the paper if the reviewers think it can be useful.

Reviewer #1: The idea to split different plasmids between different strains has been proposed and implemented by the group of the lead author in two publications (refs. 37, 40). While an argument could be made that these references dealt with combinatorial logic in which the order of inputs was engineered not to matter, the technical difference in terms of construct design is minor and the main difference is in the "optics" and the interpretation of essentially the same technical approach.

Answer: Reference 40 (Guiziou et al., 2018) is a theoretical paper describing how to implement combinatorial logic circuits in multicellular systems. The experimental implementation is provided in ref 37 (Guiziou et al., 2019).

The current paper builds of course on this principle and takes advantage of distributing the computational labor within a multicellular system. Now, our work is clearly implementing a different kind of logic, so we do not really get the point of getting an "essentially same technical approach"-would the same thing could not be said about papers using orthogonal repressors libraries that have been used to encode Boolean (12) and sequential logic (13)? Yet they clearly have different applications and both are very useful.

Here we apply distributed computation to implement Boolean and history-dependent logic, and in both cases take advantage of the unique features offered by this system. Nothing wrong with that. As a technical point, in ref 37 (Guiziou et al., 2019) we characterized our Boolean logic devices using plasmids constitutively expressing one or multiple integrases in place of inputs. This did not matter in this case as the input order was not relevant. In this article, we used inducible integrase devices, so that we could apply inputs in different orders

Reviewer #1: 2. Application to tasks in molecular logic

The authors choose a very special framework to interpret their transitions as "lineages". This implies that inputs are not applied more than once (and if they do, there is no effect because the irreversible switch acts the first time the recombinase is activated).

Answer: The reviewer raises a good point. The framework described by the reviewer is actually the same used in other papers of the field (3, 5). It indeed describes a particular case of sequential logic in which inputs cannot occur at the same time (fundamental mode (14) and cannot affect the system more than once. As inputs are applied, the system gradually transitions into narrower "branches" of a decision tree. Here we called these branches "lineages" as this is a more relevant language for biology.

The systems based on permanent recombination reactions are by nature "one-shot". Which means, you can only transition between states irreversibly. There are indeed other kinds of sequential logic systems, like those in which inputs can act multiple times. The more evident examples are binary counters. To do so, the rewritability of the memory system is required. An

impressive example of scaling such design for sequential logic is the paper by Andrews et.(13), based on mutual inhibition design (15, 16).

Recombinase systems have distinct features, which are complementary: their genetic footprint is smaller, memory storage does not require feedback, reducing the metabolic load. And as the state is physically stored within the molecular structure of DNA, these systems give access to different readout modalities, even if the cells die. In all, these are different kinds of devices, with different applications, they can actually be used together (17) and while we are aware of the limitations highlighted by the reviewer, all systems have their limitations, and recombinase sequential logic can still be tremendously useful.

In addition, we and others (18–20) have shown that using recombination directionality factors one could toggle the DNA between two orientations. We agree that building such a family of devices, matching their input/output levels, and connecting them is a significant amount of work, but nonetheless it is conceivable to do so in the future and would be an important avenue to pursue. Such systems would allow the response to the same input multiple times. We address these points in the discussion.

Reviewer #1: Their lineages are really sequences of inputs of the following format: "first appearance of a happens before the first appearance of b, that happens before the first appearance of c...". In this sense, for N inputs there are N! sequences, as correctly stated by the authors; N! behaves exponentially for large Ns. With that, the number of outputs the authors can generate is not N! but rather on the order of N; moreover, for some sequences of inputs, the output is triggered in the middle of a sequence. So my question is, what exactly is the computational power of this approach? I agree that it can be engineered to generate a readout when a particular sequence of inputs is provided. This however was shown by Friedland et al. With strain mixing, an external observer can detect up to N different sequences out of N!, therefore, most sequences would result in no output.

Answer: Our design framework supports up to $\sum_{k=0}^N \frac{N!}{k!}$ outputs (with N the number of inputs), such as a different output per state, for details please see methods, *Automated generation of genetic designs*, page 38. (N! corresponds to the number of different lineages and as well to the maximum number of cells needed to implement any N-input history-dependent program.)

The design of programs with large numbers of outputs can be obtained in the CALIN web-interface. Our design framework allows theoretically the implementation of all history-dependent logic circuits with up to N input, corresponding to $\sigma = \sum_{k=0}^N \frac{N!}{k!}$ different states, and with a maximum of one different output for each input state.

The computation power, defined here as the number of logic circuits implementable, is of up to $5 \cdot 10^{19}$ programs for 3 inputs and 16 outputs, being the maximum of outputs, and $2 \cdot 10^{18}$ programs for 4 inputs and 65 different outputs (Fig. S21). Experimentally, we implemented a set of logic circuits that we believe are representative of the different types of logic circuits that can be implemented through our framework. We used here as outputs 3 different fluorescent proteins that are straightforward to detect, facilitating precise circuit characterization, as our objective was to provide to the community tools to construct in a straightforward manner history-dependent programs.

Having 64 different outputs will be a challenge especially if the outputs are reporter proteins. But we could easily increase the number of outputs using combinations of fluorescent proteins and other types of reporters such as pigments. And if implemented in eukaryotes we could also use specific localization of fluorescent protein to increase the number of readouts. As well the state can be determined by sequencing such as in Roquet et al.

More importantly, and here we want to make this point really clear: our devices are not only meant to provide a visible output (detector) but also to control the expression of specific genes like transcription factors, enzymes, or drugs. In that sense, these devices could be used in other contexts than biosensing, for example in therapeutic bacteria. We provide examples in the introduction.

To fully answer to the reviewer, in a single cell, a logic device has $N+1$ different states and therefore $N+1$ different outputs are possible, giving the number of programs for a single-cell device of 256, for 3 inputs and 3125, for 4 inputs ($(N+1)^{(N+1)}$).

As correctly stated by the reviewer, “for some sequences of inputs, the output is triggered in the middle of a sequence”, this characteristic is intentional and desirable. We believe that it can be of interest to have a circuit which allows expression of a specific marker in the case of input a then input b have been present but not input c. As an example, for the study of genetic networks involved in morphogenesis, intermediate states of development could be tracked using this type of circuits. In general, our approach could help solve basic research questions in different species, for instance by localizing the activity of particular promoters by detecting fluorescent proteins in a range of history-dependent states. Additionally, using other types of genes of interest, as genes involved in motility or adhesion, our scaffold can control the expression of different phenotypic responses. The control of these phenotypic responses in a history-dependent manner allows programming the bacterial behavior to control biological processes such as gut or tumor colonization.

Reviewer #1: The extension to multiple strains does not strike me as a big conceptual leap; focusing on a single strain, the main result of the paper, if I rephrase it without undue embellishments, is as follows:

"For a pre-defined sequence of chemical inputs, one can engineer a DNA cassette that will generate GFP if and only if this particular sequence of inputs is provided to the strain harboring the cassette".

This would be a meaningful statement to make, however I could not find explicit reference to this result in the manuscript.

Answer: Here we might need to clarify a few points that maybe we have not presented in a sufficiently clear manner. First, the definition provided here by the reviewer is a bit restrictive. For a given sequence of input (in a single strain), our system can express one or multiple genes (fluorescent or else, see below) in various states. In addition, the history-dependent programs are executed at the multicellular level, and in fact, as a population, our system can respond to any sequence of input and produce outputs in any state. We rewrote the introduction to better describe how the systems work:

"We designed modular, standard DNA scaffolds in which genes can be inserted at specific locations that become transcriptionally active in different recombination states. Scaffolds allow the implementation in one strain of all possible gene-expression programs for a specific sequence of inputs. Programs requiring responses for different sequences of inputs are implemented by composing multiple strains. "

A more detailed explanation is provided in the results section.

Second, to complete the first part, we have to disagree with the reviewer about the importance of the multicellular nature of the system. Distributing the computational labor between strains can solve several problems. It allows scaffold standardization, parts reuse, lower metabolic burden, all of these contributing to improving the reliability of the engineered system. This is well exemplified by robust execution of history-dependent programs over 4 days, with a high robustness as quantified in Fig. 7. In addition, multicellular systems are highly modular and composable, and this allows execution of many programs using a very reduced number of logic circuits that can be composed differently, and facilitate design automation and overall predictability of the engineered system.

We have modified the introduction to make these points more clear.

Reviewer #1: Indeed, the examination of their three-input plasmids revealed that the same output could appear in different branches of the tree and at different time points. This is a serious problem, because it would mean that the same output represents different sequences of inputs, and thus the latter cannot be deduced from the output alone.

Answer: Here again and to be clear, the scaffold is not limited to expressing fluorescent reporters. The use of fluorescent output allowed us to characterize the scaffold's operation and

optimize them. We used 3 different fluorescent reporter proteins (BFP, GFP, RFP) but in fact, we could use a larger number of reporter proteins, as we explained above.

I think one point we want to clarify with the reviewer is that our system is not just a reporter system. It not only can be used to deduce that a given number and sequence of inputs occurred, but also to control the expression of genes of interest for several applications, including therapeutics and material engineering. In this case, there would be situations in which expressing the same output in different states would be desirable.

Regarding the use as a reporter, we describe in figure one the types of programs (multiple outputs, single output) and their applications. Single output in various states could still be useful, for example, to know if a particular input appeared (independently of its order in the sequence), or if cells have entered a specific lineage. We agree that this is partial, but nevertheless useful information in some cases. In addition, other modalities are described here and in the discussion to increase the multiplexing capabilities of reporter outputs.

Reviewer #1: Computational design of 4- and 5- input sequences was described but the cassettes have not been tested.

Answer: This is true, we have not yet tested the 4- and 5-input scaffolds. We believe that with the robustness of serine integrases and our optimization framework OSIRiS the construction of the 4- and 5-input scaffolds will be straightforward.

Reviewer #1: To summarize, I think the paper could be of potential interest to Nature Communications but it has to be thoroughly rewritten, with the focus on the novelty and the computation power of the approach (what exactly does it compute? Does it always work within the boundaries? Are there "undetectable" sequences?).

We have thoroughly rewritten the paper, including the introduction and the result section to address the reviewer's comments. We believe the paper has been improved by those comments.

The aspect of multiple strains working in parallel should be reduced to its correct proportions, as a trivial way to run a number of detectors, and not as some major conceptual breakthrough.

Answer: The use of multiple strains is not a "trivial" way to run multiple detectors in parallel. Distributing program execution between strains allows for (i) the use of a limited number of logic gates that can be differentially combined to produce all logic functions. (ii) the reuse of well optimized and standardized biological components in the different strains, leading to reliable systems with predictable behavior as we measure in quantifying robustness. (iii) Because of this modular approach, we can automate circuit design in a straightforward manner.

Reviewer #1: The focus should be on a single unique sequence detector, on the generality of the design approach (e.g. does a unique solution exist for any arbitrary sequence of inputs, which will respond exclusively to this sequence), the scalability beyond 4 and 5 inputs.

Answer: Our systems are actually designed to be programmable not to respond to unique sequences of inputs, but to multiple numbers of inputs given in different sequences. Each sequence being detected by one strain, and this strain being capable of producing a different output depending on which input appeared or not in the sequence. Regarding the scalability beyond 4 or 5 input, we added a note in the discussion. In summary, we believe that the really high number of strains required to scale beyond 5-input would require changes in the design or integration between biological and physical devices. These are actually interesting avenues to be pursued.

Reviewer #1: An experimental demonstration of 4- or 5-step sequence would certainly add to the paper.

Answer: We agree with the reviewer that it would be nice to provide such experiments in the future. We believe in this paper we already demonstrate the robustness of our approach.

Reviewer #1: The degree to which the authors would address these concerns will affect my eventual evaluation of the manuscript one way or the other. For now too many things are left open. The already built in non-uniqueness, with the same output appearing in many different tree branches, casts serious doubts on the above raised points.

Answer: We have clarified the comment about the same output appearing multiple times in the paragraphs above. We have thoroughly rewritten the paper to address the reviewer's comments. We thank the reviewers for these comments that in our view have improved the quality of the paper. We hope the reviewers will find the changes made now the paper suitable for publication in Nature Communications.

Reviewer #2 (Remarks to the Author):

In this paper Zúñiga et al. present a very interesting methodology for developing genetic circuits able to produce history-dependent responses. Despite this work can be of interest for the scientific community, there are several questions that must be addressed previous publication.

1. The method proposed is based on the use of recombinases to induce irreversible DNA modifications. Because experiments were done at population level, which is the efficiency of

recombination? In other words, for a given population how many cells do not perform the proper DNA modification?

This is a very relevant aspect because the sequential expression of different recombinases with different efficiencies can induce an accumulative effect. As a consequence, at the end of the experiment could be a distribution of populations expressing different genes. For simplicity, assuming that all recombinases have the same recombination efficiency p , after N inputs only p^N fraction of the final population will exhibit the correct behaviour. For instance, assuming that $p=0.85$, i.e. 85% of cells properly recombine, for systems responding to $N=4$ inputs only 52% of the cells would accumulate the right sequence of DNA modifications. Obviously, this effect will be more relevant for systems responding to a higher number of inputs. Is this effect observed in the experiments? This point has to be addressed in the paper.

Answer: Thanks for your comment, in fact, we realized we could have mentioned the recombinase efficiency more clearly. We indeed quantified the recombinase efficiency and called it “switching rate”; we provide tables of the percentage of cells in each state in supplementary data. Individual recombinase switching efficiencies are of ~92% (TP901-1), ~97% (Int5), and ~98% (Bxb1). These values have been determined using specific DNA states as targets and of course, recombination efficiency can also be affected by specific states of the DNA sequence. But still, because we used OSIRIS to synthesize intermediate states, the targets are the same states that in the full system, and we believe these values are pretty accurate.

The reviewer makes a good point, if recombination efficiency is too low, then after a sequential reaction, we would end up with a lower proportion of cells producing the correct output. According to the individual recombinase switching efficiency stated previously, the theoretical recombinase efficiency after the 3 integrases action should be of 87%. We indeed observe similar experimental results for the 3 input programs. We provide tables with the percentage of recombined cells in every state of the different implemented programs in Fig. S4 and Fig. S12, from the population gated for each fluorescent protein (GFP, RFP, and BFP), using Flow-cytometry data. From these tables, we can observe that after 3 sequential recombinations the percentage of cells expressing the expected fluorescent proteins range from 92% to 73.5% with a mean of 84% (see table for 3SP5 program in Fig. S12). Theoretically, in the case that a fourth recombinase will have a switching rate similar to the one of TP901-1, the proportion of cells properly recombined for $N=4$ inputs would be 80%.

Following the reviewer's comment, we decided to add the recombinase efficiency in Fig. S9 for each recombinase, evaluated in the scaffold for 3-input history-dependent programs. These values were obtained under the growth conditions described on methods. In addition, we added a comment in the discussion to precise the importance of optimizing recombinase switching rate for the engineering of robust systems.

Reviewer #2:

2. Related with the previous question, authors claim “Because all programs had a high switching rate, we extended...”, However, “high switching rate” is not very precise, can these rate be quantified?

Answer: Please see the answer above.

Reviewer #2:

3. Authors claim “For a given number of inputs, the maximum number of strains needed is equal to the number of lineages ($N!$ for N inputs). However, most functions are implementable with fewer than the maximum number of strains, as the number of strains depends on the number of lineages in which gene expression is required.” The expression “with fewer” is ambiguous. A chart representing the number of functions versus the number of strains for different numbers of inputs would be very useful.

Answer: To provide the information related to the maximum number of strains needed we plotted the distribution of the number of the strains required for the implementation of all history dependent programs, in particular for 3 -input 1-output history-dependent programs, please see Fig S1. The median is 4.6 strains.

Reviewer #2:

4. Considering ON and OFF gene expression as different states, for N inputs should be $N! \cdot 2^N$ combinations, i.e. $N!$ branches with 2^N possible configurations of ON-OFF genes. Figure 2 shows nine different history-dependent responses to two inputs. Are these circuits arbitrary examples? Why these cases are selected?

Answer: Good point. The nine different history-dependent programs represented in figure 2 were not chosen arbitrarily, they were chosen to be representative of all 2-input single-cell history-dependent programs. We have updated the text to make this clearer.

In Figure 2 we show:

1. Programs 2SP1 to 2SP4, which are strictly history-dependent, single-cell 2-input programs with ON gene expression in one, two or all states of the lineage b then a. are and are using a single reporter output (GFP).
2. Programs 2SP6 to 2SP9 are strictly history-dependent, single-cell 2-input programs with ON gene expression in one, two or every state of the lineage a then b, and have as output the three different reporter genes BFP, RFP, and GFP.

While even combinatorial logic programs are implementable with this design, we didn't characterize them. These programs are also implementable based on the design method of ref37 and 40 which is simplest: for example, the logic function $\text{not}(a).\text{not}(b)$, $\text{not}(a).b$, $b.\text{not}(a)$ which are implementable in one lineage following the design presented here and could have been shown in figure 2. The only exception we made was for the program 2SP5 which is indeed the Boolean logic function $b.\text{not}(a)$ implemented using our design. We decided to show that device because we use it afterward for the implementation of 2-input multi-lineage programs.

Reviewer #2:

5. Related to the previous question, creation of multicellular systems able to access to all $N! \cdot 2^N$ combinations could need more of N recombinases in each strain? Using $2N$ recombinases all ON-OFF combinations are accessible, but probably this can be done with less recombinases. How the number of recombinases limits the subset of accessible functions?

Answer: Using N recombinases, all ON and OFF combinations for N inputs are accessible. Reducing the number of recombinases to Y (with $Y < N$) will limit the set of accessible functions to functions with Y inputs.

Reviewer #2:

6. Is there leakiness in the expression of the recombinases under inducible promoters? Has RBS strength been used to minimize leakiness? Could be useful to present genetic parts sequence as supplementary information.

Answer: All sequences with genetic parts are available in an excel document in supplementary information. Additionally, all the sequences and plasmid of this work will be available from Addgene (pITC, integrase triple controller, ID: 126540).

The 2-input history-dependent programs were implemented using the dual controller plasmid, with recombinases Bxb1 and Tp901.1 under the inducible promoter pBAD and pTET, respectively. This plasmid was optimized previously in Bonnet et al 2013. The sequence is available on Addgene website, Dual-recombinase-controller (plasmid #44456).

The 3-input history-dependent programs were implemented using the triple controller plasmid, constructed and optimized in this work, adding to the dual controller a third recombinase (Integrase 5) under the inducible promoter pBEN. As a first step, we optimized the expression of BenR using different promoters and RBS. Then we optimized the expression and recombinase rate of this construct by creating a library with different RBS and ssRA degradation tags. And schematics of this optimization is shown in Fig. S19, including the genetic parts for each construct.

Reviewer #2:

7. How the use of GFP variants instead of using different reporters can provide confusing results? In particular, related with recombination efficiency, if populations subsets that have not

properly recombined express a different GFP than cells that have recombined right, can this conceal this fact?

Answer: Our objective of using GFP variants was to show that we are able to implement programs with single output in addition to programs with different outputs. For some applications, the expression of an enzyme for the production of specific drugs will be required at different states, by characterizing these circuits our aim was to show that this was possible. All programs implemented using GFP variants can definitely be implemented using different reporters. In fact, in our characterization the GFP being different we observe different GFP intensity in the different states corresponding to the expression of the different GFP variants which confirm the fact that these devices are in the correct recombination state (for details about fold change please see Fig. S5) for details about fluorescence intensity please see FCS files links for each program available on Flow repository in methods section.

Reviewer #2:

8. It is surprising the stability of mixed populations growth without observable competition effects. However, when the number of branches increases the number of different strains coexisting increases as well. Can this compromise the stability of circuits? For instance, for systems with 3 inputs could be necessary the coexistence of 6 different strains in the same culture, and scale up to 24 for 4 inputs. How can this compromise the applicability of these circuits for systems beyond 2 inputs?

Answer: Indeed, systems with a large number of strains can be challenging. In this paper, we tested circuits for up to 3 strains, and all circuits tested behaved as expected, being really stable. We did not observe issues with cultures of 3 strains growing during 4 days, in fact, these cultures were stable and led to robust outputs. We believe that the fact that recombinase memory maintenance does not require feedback reduces metabolic burden and avoids cell differences in growth rates.

For N inputs, the number of strains ranges from 1 to N!, therefore this design is scalable for sure to up to 5 inputs and 3 strains as it is the highest number of strains that we have tested to grow together. Additionally, we provide evidence that using Boolean logic devices we can minimize the number of devices thus the number of strains used to implement history-dependent programs, as we show in Fig. S15.

We know that multiple strain implementation could be a challenge in our system, but in fact, other strategies can be used to address this challenge. In this sense, cell-cell communication could be useful to control and maintain strains proportion within the population over time (21). Of course, other options such as introducing metabolic dependencies or metabolite overproduction in strains used to implement multicellularity are available, as recently proposed by (9) and (10). Finally, specific devices could be designed to maintain cells physically separated (11). All these points are addressed in the discussion section of the paper.

Reviewer #2:

9. According to the methodology, this system cannot be applied for temporal input patterns in which the same input appears more than one time, for instance in oscillatory patterns. How can this aspect limit the range of potential applications of the methodology presented? and can be possible to extend the approach to include patterns with repeated inputs? I sort discussion about this limitation can be useful

Answer: Reviewer #1 had the same comment, please see answer p6. Systems with rewritable memory are needed especially in the case of oscillators (think cell cycle counter...). Also, additional logic gates would be needed so that the system is kept “locked” and unresponsive until the input has disappeared (to avoid overcounting). Andrews et al. nicely paved the way for such systems (13). We have modified the discussion to address this point.

References

1. B. H. Weinberg, N. T. H. Pham, L. D. Caraballo, T. Lozanoski, A. Engel, S. Bhatia, W. W. Wong, Large-scale design of robust genetic circuits with multiple inputs and outputs for mammalian cells. *Nat. Biotechnol.* **35**, 453–462 (2017).
2. A. J. Podhajski, N. Hasan, W. Szybalski, Control of cloned gene expression by promoter inversion in vivo: construction of the heat-pulse-activated att-nutLp-att-N. *Gene.* **43**, 325 (1986).
3. T. S. Ham, S. K. Lee, J. D. Keasling, A. P. Arkin, Design and Construction of a Double Inversion Recombination Switch for Heritable Sequential Genetic Memory. *PLoS One.* **3**, e2815 (2008).
4. A. E. Friedland, T. K. Lu, X. Wang, D. Shi, G. Church, J. J. Collins, Synthetic Gene Networks That Count. *Science.* **324**, 1199–1202 (2009).
5. N. Roquet, A. P. Soleimany, A. C. Ferris, S. Aaronson, T. K. Lu, Synthetic recombinase-based state machines in living cells. *Science.* **353**, aad8559 (2016).
6. V. Hsiao, Y. Hori, P. W. Rothemund, R. M. Murray, A population-based temporal logic gate for timing and recording chemical events. *Mol. Syst. Biol.* **12**, 869 (2016).
7. S. D. Colloms, C. A. Merrick, F. J. Olorunniji, W. M. Stark, M. C. M. Smith, A. Osbourn, J. D. Keasling, S. J. Rosser, Rapid metabolic pathway assembly and modification using serine integrase site-specific recombination. *Nucleic Acids Res.* **42**, e23 (2014).
8. K.-H. K. Chow, M. W. Budde, A. A. Granados, M. Cabrera, S. Yoon, S. Cho, T.-H. Huang, N. Koulana, K. L. Frieda, L. Cai, C. Lois, M. B. Elowitz, Imaging cell lineage with a synthetic digital recording system. *bioRxiv* (2020), p. 2020.02.21.958678.

9. W. Kong, D. R. Meldgin, J. J. Collins, T. Lu, Designing microbial consortia with defined social interactions. *Nat. Chem. Biol.* (2018), doi:10.1038/s41589-018-0091-7.
10. M. Ziesack, T. Gibson, J. K. W. Oliver, A. M. Shumaker, B. B. Hsu, D. T. Riglar, T. W. Giessen, N. V. DiBenedetto, L. Bry, J. C. Way, P. A. Silver, G. K. Gerber, Engineered Interspecies Amino Acid Cross-Feeding Increases Population Evenness in a Synthetic Bacterial Consortium. *mSystems*. **4** (2019), doi:10.1128/mSystems.00352-19.
11. J. Macia, R. Manzoni, N. Conde, A. Urrios, E. de Nadal, R. Solé, F. Posas, Implementation of Complex Biological Logic Circuits Using Spatially Distributed Multicellular Consortia. *PLoS Comput. Biol.* **12**, e1004685 (2016).
12. A. A. K. Nielsen, B. S. Der, J. Shin, P. Vaidyanathan, V. Paralanov, E. A. Strychalski, D. Ross, D. Densmore, C. A. Voigt, Genetic circuit design automation. *Science*. **352**, aac7341 (2016).
13. L. B. Andrews, A. A. K. Nielsen, C. A. Voigt, Cellular checkpoint control using programmable sequential logic. *Science*. **361** (2018), doi:10.1126/science.aap8987.
14. F. J. Hill, G. R. Peterson, *Introduction to Switching Theory and Logical Design* (philpapers.org, 1968).
15. Z. Toman, C. Dambly-Chaudière, L. Tenenbaum, M. Radman, A system for detection of genetic and epigenetic alterations in *Escherichia coli* induced by DNA-damaging agents. *J. Mol. Biol.* **186**, 97–105 (1985).
16. T. S. Gardner, C. R. Cantor, J. J. Collins, Construction of a genetic toggle switch in *Escherichia coli*. *Nature*. **403**, 339–342 (2000).
17. L. Yang, A. A. K. Nielsen, J. Fernandez-Rodriguez, C. J. McClune, M. T. Laub, T. K. Lu, C. A. Voigt, Permanent genetic memory with >1-byte capacity. *Nat. Methods*. **11**, 1261–1266 (2014).
18. J. Bonnet, P. Subsoontorn, D. Endy, Rewritable digital data storage in live cells via engineered control of recombination directionality. *Proc. Natl. Acad. Sci. U. S. A.* **109**, 8884–8889 (2012).
19. J. M. Bernabé-Orts, A. Quijano-Rubio, M. Vazquez-Vilar, J. Mancheño-Bonillo, V. Moles-Casas, S. Selma, S. Gianoglio, A. Granell, D. Orzaez, A memory switch for plant synthetic biology based on the phage ϕ C31 integration system. *Nucleic Acids Res.* (2020), doi:10.1093/nar/gkaa104.
20. F. J. Olorunniji, A. L. McPherson, S. J. Rosser, M. C. M. Smith, S. D. Colloms, W. M. Stark, Control of serine integrase recombination directionality by fusion with the directionality factor. *Nucleic Acids Res.* (2017), doi:10.1093/nar/gkx567.
21. K. Brenner, L. You, F. H. Arnold, Engineering microbial consortia: a new frontier in synthetic biology. *Trends Biotechnol.* **26**, 483–489 (2008).

ANNEX: Comparison of the characteristics of various history-dependent logic systems operating in living cells.

Publication	Biochemical mechanism used	Type of logic implemented	N° of inputs theoretical/implemented	N° of GOI outputs (DNA states?) implemented	Completeness	Automated design frameworks?	N° of experimentally implemented programs	N° of strains
(Ham et al. , 2008)	Serine integrases	Order-of-occurrence of event, irreversible	2 / 2	No output gene, and 4 different DNA states	No	No	1 two-input program, not fully working	1
(Friedland et al. , 2009)	Tyrosine integrases	2 circuits design for pulse tracking, one for order-of-occurrence of event which is not fully irreversible	3 / 3	One output gene, and 3 different DNA states.	No	No	1 three-input program	1
(Lou et al. , 2010)	Repressors	Push-on push-off switch	1 / 1	2 output genes, 1 DNA states	No	No	1 one-input program	1
(Zhang et al. , 2014)	Repressors, transcription factors	Reversible sequential logic, Pavlovian-like conditioning circuit	2/2	2 output genes	No	No	1 two-input program	1
(Urrios et al. , 2016)	Repressors	One memory switch - double-negative feedback with memory	2/2	1 output gene	No	No	1 two-input program	Multiple strains (3 strains)
(Roquet et al. , 2016)	Serine integrases	Order-of-occurrence, irreversible	7 / 3	Different DNA state per input state (16 for 3-input), 3 different output genes and 5 different output states (2 combining the simultaneous expression of 2 output genes).	Completeness for DNA as output, not specified for gene expression as output.	Yes	2 two-input programs 3 three-input programs Total: 5	1
(Sheth et al. , 2017)	CRISPR-Cas9	Order of occurrence, irreversible	3/3	No output gene-state encoded at the DNA level, probabilistic system	Complete at the DNA level for up to 3 inputs, no gene expression as output.	No	16 Three-input programs tested, 50% fully functional	Detected at the population level
(Andrews et al. , 2018)	Repressors	Reversible sequential logic composed of multiple latches	4 / 4	Up to 6 outputs and each output is detected separately, as each strain contains a reporter for one output.	Not specified.	No	1 four-input program, 7 three-input programs Total: 8	1
This work	Serine integrases	Order of occurrence, irreversible	5 / 3	4 output genes, at the population level, one DNA state per input state.	Complete to 5 inputs	Yes	11 two input programs, 10 Three-input programs Total: 21	Multiple strains

REFERENCES

Andrews, L. B., Nielsen, A. A. K., & Voigt, C. A. (2018). Cellular checkpoint control using programmable sequential logic. *Science*, 361(6408). <https://doi.org/10.1126/science.aap8987>

Friedland, A. E., Lu, T. K., Wang, X., Shi, D., Church, G., & Collins, J. J. (2009). Synthetic gene networks that count. *Science*, 324(5931), 1199–1202.

Ham, T. S., Lee, S. K., Keasling, J. D., & Arkin, A. P. (2008). Design and construction of a double inversion recombination switch for heritable sequential genetic memory. *PLoS One*, 3(7), e2815.

- Lou, C., Liu, X., Ni, M., Huang, Y., Huang, Q., Huang, L., Jiang, L., Lu, D., Wang, M., Liu, C., Chen, D., Chen, C., Chen, X., Yang, L., Ma, H., Chen, J., & Ouyang, Q. (2010). Synthesizing a novel genetic sequential logic circuit: a push-on push-off switch. *Molecular Systems Biology*, 6, 1–11.
- Roquet, N., Soleimany, A. P., Ferris, A. C., Aaronson, S., & Lu, T. K. (2016). Synthetic recombinase-based state machines in living cells. *Science*, 353(6297), aad8559.
- Sheth, R. U., Yim, S. S., Wu, F. L., & Wang, H. H. (2017). Multiplex recording of cellular events over time on CRISPR biological tape. *Science*, 358(6369), 1457–1461.
- Urrios, A., Macia, J., Manzoni, R., Conde, N., Bonforti, A., de Nadal, E., Posas, F., & Solé, R. (2016). A Synthetic Multicellular Memory Device. *ACS Synthetic Biology*, 5(8), 862–873.
- Zhang, H., Lin, M., Shi, H., Ji, W., Huang, L., Zhang, X., Shen, S., Gao, R., Wu, S., Tian, C., Yang, Z., Zhang, G., He, S., Wang, H., Saw, T., Chen, Y., & Ouyang, Q. (2014). Programming a Pavlovian-like conditioning circuit in *Escherichia coli*. *Nature Communications*, 5, 3102.

Reviewers' Comments:

Reviewer #1:

Remarks to the Author:

The authors have by and large addressed my concerns about the manuscript; a manuscript has been improved as a result and can be published by Nature Communications. They should, however, cite and discuss additional papers dealing with recombinase in synthetic biology: PMID: 25306443 and PMID: 29133926.

Reviewer #2:

Remarks to the Author:

Authors have properly addressed all questions. In my opinion this paper is ready for publication.

REVIEWERS' COMMENTS:

Reviewer #1 (Remarks to the Author):

The authors have by and large addressed my concerns about the manuscript; a manuscript has been improved as a result and can be published by Nature Communications. They should, however, cite and discuss additional papers dealing with recombinase in synthetic biology: PMID: 25306443 and PMID: 29133926.

Reviewer #2 (Remarks to the Author):

Authors have properly addressed all questions. In my opinion this paper is ready for publication.

Javier Macía Santamaría, PhD.

RESPONSE:

We are extremely happy that the reviewers find the paper is now suitable for publication in Nature Communications. We'd like to thank them again for their time and comments that have clearly improved the quality of the article.

Regarding reviewer's #1 citation advice, we know these papers and thank the reviewer for the suggestion. They are nice, solid work but we believe that they are a bit far from the specific topic of our paper and that the references we cite already cover a wide range of the literature- so we have kept our references list unchanged. We hope the reviewer will understand.